# Koopman Spectrum Nonlinear Regulators and Efficient Online Learning

**Motoya Ohnishi**                                          *mohnishi@cs.washington.edu*
*Paul G. Allen School of Computer Science & Engineering*
*University of Washington*

**Isao Ishikawa**                                          *ishikawa.isao.zx@ehime-u.ac.jp*
*Ehime University*
*RIKEN Center for Advanced Intelligence Project*

**Kendall Lowrey**                                          *kendall.lowrey@gmail.com*
*Et Cetera Robotics*

**Masahiro Ikeda**                                          *masahiro.ikeda@riken.jp*
*RIKEN Center for Advanced Intelligence Project*
*Keio University*

**Sham Kakade**                                          *sham@seas.harvard.edu*
*Harvard University*

**Yoshinobu Kawahara**                                          *kawahara@ist.osaka-u.ac.jp*
*Graduate School of Information Science and Technology, Osaka University*
*RIKEN Center for Advanced Intelligence Project*

**Reviewed on OpenReview:** *https://openreview.net/forum?id=thfoUZugvS*

## Abstract

Most modern reinforcement learning algorithms optimize a cumulative single-step cost along a trajectory. The optimized motions are often 'unnatural', representing, for example, behaviors with sudden accelerations that waste energy and lack predictability. In this work, we present a novel paradigm of controlling nonlinear systems via the minimization of the *Koopman spectrum cost*: a cost over the Koopman operator of the controlled dynamics. This induces a broader class of dynamical behaviors that evolve over stable manifolds such as nonlinear oscillators, closed loops, and smooth movements. We demonstrate that some dynamics characterizations that are not possible with a cumulative cost are feasible in this paradigm, which generalizes the classical eigenstructure and pole assignments to nonlinear decision making. Moreover, we present a sample efficient online learning algorithm for our problem that enjoys a sub-linear regret bound under some structural assumptions.

## 1 Introduction

Reinforcement learning (RL) has been successfully applied to diverse domains, such as robot control (Kober et al. (2013); Todorov et al. (2012); Ibarz et al. (2021)) and playing video games (Mnih et al. (2013; 2015)). Most modern RL problems modeled as Markov decision processes consider an immediate (single-step) cost (reward) that accumulates over a certain time horizon to encode tasks of interest. Although such a cost can encode any single realizable dynamics, which is central to inverse RL problems (Ng & Russell (2000)), the generated motions often exhibit undesirable properties such as high jerk, sudden accelerations that waste energy, and unpredictability. Intuitively, the motion specified by the task-oriented cumulative cost formulation may ignore "how to" achieve the task unless careful design of cumulative cost is in place,

necessitating a systematic approach that effectively regularizes or *constrains* the dynamics to guarantee predictable global property such as stability.

Meanwhile, many dynamic phenomena found in nature are known to be represented as simple trajectories, such as nonlinear oscillators, on low-dimensional manifolds embedded in high-dimensional spaces that we observe (Strogatz, 2018). Its mathematical concept is known as *phase reduction* (Winfree, 2001; Nakao, 2017), and recently its connection to the Koopman operator has been attracted much attention in response to the growing abundance of measurement data and the lack of known governing equations for many systems of interest (Koopman, 1931; Mezić, 2005; Kutz et al., 2016).

In this work, we present a novel paradigm of controlling nonlinear systems based on the spectrum of the Koopman operator. To this end, we exploit the recent theoretical and practical developments of the Koopman operators (Koopman, 1931; Mezić, 2005), and propose the *Koopman spectrum cost* as the cost over the Koopman operator of controlled dynamics, defining a preference of the dynamical system in the reduced phase space. The Koopman operator, also known as the composition operator, is a linear operator over an observable space of a (potentially nonlinear) dynamical system, and is used to extract global properties of the dynamics such as its dominating modes and eigenspectrum through spectral decomposition. Controlling nonlinear systems via the minimization of the Koopman spectrum cost induces a broader class of dynamical behaviors such as nonlinear oscillators, closed loops, and smooth movements.

Although working in the spectrum (or frequency) domain has been standard in the control community (e.g. Andry et al. (1983); Hemati & Yao (2017)), the use of the Koopman spectrum cost together with function approximation and learning techniques enables us to generate rich class of dynamics evolving over stable manifolds (cf. Strogatz (2018)).

**Our contributions.**   The contributions of this work are three folds: First, we propose the Koopman spectrum cost that complements the (cumulative) single-step cost for nonlinear control. Our problem, which we refer to as Koopman Spectrum Nonlinear Regulator (KSNR), is to find an optimal parameter (e.g. policy parameter) that leads to a dynamical system associated to the Koopman operator that minimizes the sum of both the Koopman spectrum cost and the cumulative cost. Note that "Regulator" in KSNR means not only *controller* in control problems but a more broad sense of regularization of dynamical systems for attaining specific characteristics. Second, we show that KSNR paradigm effectively encodes/imitates some desirable agent dynamics such as limit cycles, stable loops, and smooth movements. Note that, when the underlying agent dynamics is known, KSNR may be approximately solved by extending any nonlinear planning heuristics, including population based methods. Lastly, we present a (theoretical) learning algorithm for online KSNR, which attains the sub-linear regret bound (under certain condition, of order $\tilde{O}(\sqrt{T})$). Our algorithm (Koopman-Spectrum LC$^3$ (KS-LC$^3$)) is a modification of Lower Confidence-based Continuous Control (LC$^3$) (Kakade et al., 2020) to KSNR problems with several technical contributions. We need structural assumptions on the model to simultaneously deal with the Koopman spectrum cost and cumulative cost. Additionally, we present a certain Hölder condition of the Koopman spectrum cost that makes regret analysis tractable for some cost such as the spectral radius.

---

**Key Takeaways**

KSNR considers the *spectrum cost* that is not subsumed by a classical single-step cost or an episodic cost, and is beyond the MDP framework, which could be viewed as a generalization of eigenstructure / pole assignments to nonlinear decision making problems. The framework systematically deals with the *shaping* of behavior (e.g., ensuring stability, smoothness, and adherence to a target *mode* of behavior). Because we employ this new cost criterion, we strongly emphasize that *this work is not intended to compete against the MDP counterparts, but rather to illustrate the effectiveness of the generalization we make for systematic behavior shaping.* For online learning settings under unknown dynamics, the spectrum cost is unobservable because of inaccessible system models; it thus differs from other costs such as a policy cost. As such, we need specific structural assumptions that are irrelevant to the Kernelized Nonlinear Regulator (Kakade et al., 2020) to devise sample efficient (if not computationally efficient) algorithm. Proof techniques include some operator theoretic arguments that are novel in this context.

---

**Notation.** Throughout this paper, $\mathbb{R}$, $\mathbb{R}_{\geq 0}$, $\mathbb{N}$, $\mathbb{Z}_{>0}$, and $\mathbb{C}$ denote the set of the real numbers, the non-negative real numbers, the natural numbers ($\{0, 1, 2, \ldots\}$), the positive integers, and the complex numbers, respectively. Also, $\Pi$ is a set of dynamics parameters, and $[H] := \{0, 1, \ldots H - 1\}$ for $H \in \mathbb{Z}_{>0}$. The set of bounded linear operators from $\mathcal{A}$ to $\mathcal{B}$ is denoted by $\mathcal{L}(\mathcal{A}; \mathcal{B})$, and the adjoint of the operator $\mathscr{A}$ is denoted by $\mathscr{A}^\dagger$. We let $\det(\cdot)$ be the functional determinant. Finally, we let $\|x\|_{\mathbb{R}^d}$, $\|x\|_1$, $\|\mathscr{A}\|$, and $\|\mathscr{A}\|_{\mathrm{HS}}$ be the Euclidean norm of $x \in \mathbb{R}^d$, the 1-norm (sum of absolute values), the operator norm of $\mathscr{A} \in \mathcal{L}(\mathcal{A}; \mathcal{B})$, and the Hilbert–Schmidt norm of a Hilbert-Schmidt operator $\mathscr{A}$, respectively.

## 2 Related work

Koopman operator was first introduced in Koopman (1931); and, during the last two decades, it has gained traction, leading to the developments of theory and algorithm (e.g. Črnjarić-Žic et al. (2019); Kawahara (2016); Mauroy & Mezić (2016); Ishikawa et al. (2018); Iwata & Kawahara (2020); Burov et al. (2021)) partially due to the surge of interests in data-driven approaches. The analysis of nonlinear dynamical system with Koopman operator has been applied to control (e.g. Korda & Mezić (2018); Mauroy et al. (2020); Kaiser et al. (2021); Li et al. (2019); Korda & Mezić (2020)), using model predictive control (MPC) framework and linear quadratic regulator (LQR) although nonlinear controlled systems in general cannot be transformed to LQR problem even by lifting to a feature space. For unknown systems, active learning of Koopman operator has been proposed (Abraham & Murphey (2019)). Note our framework applied to stability regularization considers the solely different problem than that of (Mamakoukas et al., 2023). In the context of stability regularization, our framework is *not for learning the stable Koopman operator or to construct a control Lyapunov function from the learned operator* but to solve the regulator problem to balance the (possibly task-based) cumulative cost and the spectrum cost that enforces stability. We will revisit this perspective in Section 3.3.

The line of work that is most closely related to ours is the eigenstructure / pole assignments problem (e.g. Andry et al. (1983); Hemati & Yao (2017)) classically considered in the control community particularly for linear systems. In fact, in the literature on control, working in frequency domain has been standard (e.g. Pintelon & Schoukens (2012); Sabanovic & Ohnishi (2011)). These problems aim at computing a feedback policy that generates the dynamics whose eigenstructure matches the desired one; we note such problems can be naturally encoded by using the Koopman spectrum cost in our framework as well. In connection with the relation of the Koopman operator to stability, these are in principle closely related to the recent surge of interest in neural networks to learn dynamics with stability (Manek & Kolter, 2019; Takeishi & Kawahara, 2021).

In the RL community, there have been several attempts of using metrics such as mutual information for acquisitions of the *skills* under RL settings, which are often referred to as unsupervised RL (e.g. Eysenbach et al. (2018)). These work provide an approach of effectively generating desirable behaviors through compositions of skills rather than directly optimizing for tasks, but are still within cumulative (single-step) cost framework. Historically, the motor primitives investigated in, for example, (Peters & Schaal, 2008; Ijspeert et al., 2002; Stulp & Sigaud, 2013) have considered parametric nonlinear dynamics having some desirable properties such as stability, convergence to certain attractor etc., and it is related to the Koopman spectrum regularization in the sense both aim at regulating the global dynamical properties. Those primitives may be discovered by clustering (cf. Stulp et al. (2014)), learned by imitation learning (cf. Kober & Peters (2010)), and coupled with meta-parameter learning (e.g. Kober et al. (2012)).

Finally, as related to the sample efficient algorithm we propose, provably correct methods (e.g. Jiang et al. (2017); Sun et al. (2019)) have recently been developed for continuous control problems (Kakade et al., 2020; Mania et al., 2020; Simchowitz & Foster, 2020; Curi et al., 2020).

Below, we present our control framework, KSNR, in Section 3 with several illustrative numerical examples based on population based policy search (e.g. genetic algorithm), followed by an introduction of its example online learning algorithm (Section 4) with its theoretical insights on sample complexity and on reduction of the model to that of eigenstructure assignments problem as a special case. For more details about population based search that repeatedly evaluates the sampled actions to update the sampling distribution of action sequence so that the agent can achieve lower cost, see for example (Beheshti & Shamsuddin, 2013).

# 3 Koopman Spectrum Nonlinear Regulator

In this section, we present the dynamical system model and our main framework.

## 3.1 Dynamical system model

Let $\mathcal{X} \subset \mathbb{R}^{d_\mathcal{X}}$ be the state space, and $\Pi$ a set of parameters each of which corresponds to one random dynamical system (RDS) as described below. Given a parameter $\Theta \in \Pi$, let $(\Omega_\Theta, P_\Theta)$ be a probability space, where $\Omega_\Theta$ is a measurable space and $P_\Theta$ is a probability measure. Let $\mu_\Theta := (\mu_\Theta(r))_{r \in \mathbb{N}}$ be a semi-group of measure preserving measurable maps on $\Omega_\Theta$ (i.e., $\mu_\Theta(r) : \Omega_\Theta \to \Omega_\Theta$). This work studies the following nonlinear regulator (control) problem: for each parameter $\Theta \in \Pi$, the corresponding nonlinear random dynamical system is given by

$$\mathcal{F}^\Theta : \mathbb{N} \times \Omega_\Theta \times \mathcal{X} \to \mathcal{X},$$

that satisfies

$$\mathcal{F}^\Theta(0, \omega, x) = x, \quad \mathcal{F}^\Theta(r + s, \omega, x) = \mathcal{F}^\Theta(r, \mu_\Theta(s)\omega, \mathcal{F}^\Theta(s, \omega, x)), \quad \forall r, s \in \mathbb{N}, \ \omega \in \Omega_\Theta, \ x \in \mathcal{X}. \tag{3.1}$$

The above definition of random dynamical system is standard in the community studying dynamical systems (refer to (Arnold, 1998), for example). Roughly speaking, an RDS consists of the following two models:

- A model of the *noise*;

- A function representing the *physical* dynamics of the system.

RDSs subsume many practical systems including solutions to stochastic differential equations and additive-noise systems, i.e.,

$$x_{h+1} = f(x_h) + \eta_h, \quad x_0 \in \mathbb{R}^d, \ h \in [H],$$

where $f : \mathbb{R}^{d_\mathcal{X}} \to \mathbb{R}^{d_\mathcal{X}}$ represents the dynamics, and $\eta_h \in \mathbb{R}^{d_\mathcal{X}}$ is the zero-mean i.i.d. additive noise vector. Let $\Omega_0$ be a probability space of the noise, and one could consider $\Omega := \Omega_0^\mathbb{N}$ (and $\mu$ is the shift) to express the system as an RDS. Also, Markov chains can be described as an RDS by regarding them as a composition of i.i.d. random transition and by following similar arguments to the above (see Arnold (1998, Theorem 2.1.6)). As an example, consider the discrete states $\{s_1, s_2\}$ and the transition matrix

$$\text{TRANSITION MATRIX} := \begin{bmatrix} 0.8 & 0.2 \\ 0.1 & 0.9 \end{bmatrix} = 0.7 \begin{bmatrix} 1 & 0 \\ 0 & 1 \end{bmatrix} + 0.2 \begin{bmatrix} 0 & 1 \\ 0 & 1 \end{bmatrix} + 0.1 \begin{bmatrix} 1 & 0 \\ 1 & 0 \end{bmatrix},$$

where the right hand side shows a decomposition of the transition matrix to deterministic transitions. For this example, one can naturely treat this Markov chain as an RDS that generates each deterministic transition with corresponding probability at every step.

More intuitively, dynamical systems with an *invariant noise-generating mechanism* could be described as an RDS by an appropriate translation (see Figure 1 (Left) for an illustration of an RDS).

**Koopman operator** For the random dynamical systems being defined above, we define the operator-valued map $\mathcal{K}$ below, using the dynamical system model.

**Definition 3.1** (Koopman operator). Let $\mathcal{H}$ be a function space on $\mathcal{X}$ over $\mathbb{C}$ and let $\{\mathcal{F}^\Theta\}_{\Theta \in \Pi}$ be a dynamical system model. We define an operator-valued map $\mathcal{K}$ by $\mathcal{K} : \Pi \to \mathcal{L}(\mathcal{H}, \mathcal{H})$ such that for any $\Theta \in \Pi$ and $g \in \mathcal{H}$,

$$[\mathcal{K}(\Theta)g](x) := \mathbb{E}_{\Omega_\Theta}\big[g \circ \mathcal{F}^\Theta(1, \omega, x)\big], \quad x \in \mathcal{X}.$$

We will choose a suitable $\mathcal{H}$ to define the map $\mathcal{K}$, and $\mathcal{K}(\Theta)$ is the Koopman operator for $\mathcal{F}^\Theta$. Essentially, the Koopman operator represents a nonlinear dynamics as a linear (infinite-dimensional) operator that describes the evolution of *observables* in a lifted space (see Figure 1 Right).

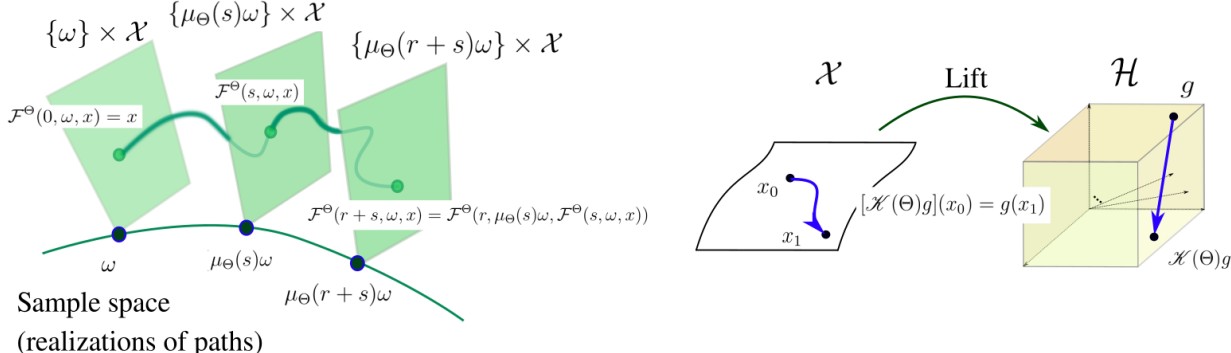

Figure 1: Left: Random dynamical system consists of a model of the *noise* and a function representing the *physical* phase space (the illustration is inspired by Arnold (1998); Ghil et al. (2008)). The RDS flows over sample space and phase space for each realization $\omega$ and for initial state $x_0$. Right: By lifting the state space to a space of observables, a nonlinear dynamical system over the state space is represented by the linear operator in a lifted space.

**Remark 3.1** (Choice of $\mathcal{H}$ and existence of $\mathscr{K}$). In an extreme (but useless) case, one could choose $\mathcal{H}$ to be a one dimensional space spanned by a constant function, and $\mathscr{K}$ can be defined. In general, the properties of the Koopman operator depend on the choice of the space on which the operator is defined. As more practical cases, if one employs a Gaussian RKHS for example, the only dynamics inducing bounded Koopman operators are those of affine (e.g. Ishikawa (2023); Ikeda et al. (2022)). However, some practical algorithms have recently been proposed for an RKHS to approximate the eigenvalues of so-called "extended" Koopman operator through some appropriate computations under certain conditions on the dynamics and on the RKHS (cf. Ishikawa et al. (2024)).

With these settings in place, we propose our framework.

### 3.2 Koopman Spectrum Nonlinear Regulator

Fix a set $X_0 := \{(x_{0,0}, H_0), (x_{0,1}, H_1), \ldots, (x_{0,N-1}, H_{N-1})\} \subset \mathcal{X} \times \mathbb{Z}_{>0}$, for $N \in \mathbb{Z}_{>0}$, and define $c : \mathcal{X} \to \mathbb{R}_{\geq 0}$ be a cost function. The Koopman Spectrum Nonlinear Regulator (KSNR), which we propose in this paper, is the following optimization problem:

$$\text{Find} \quad \Theta^\star \in \underset{\Theta \in \Pi}{\arg\min} \left\{ \Lambda[\mathscr{K}(\Theta)] + J^\Theta(X_0; c) \right\}, \tag{3.2}$$

where $\Lambda : \mathcal{L}(\mathcal{H}; \mathcal{H}) \to \mathbb{R}_{\geq 0}$ is a mapping that takes a Koopman operator as an input and returns its cost; and

$$J^\Theta(X_0; c) := \sum_{n=0}^{N-1} \mathbb{E}_{\Omega_\Theta} \left[ \sum_{h=0}^{H_n-1} c(x_{h,n}) \Big| \Theta, x_{0,n} \right],$$

where $x_{h,n}(\omega) := \mathcal{F}^\Theta(h, \omega, x_{0,n})$. In control problem setting, the problem (3.2) can be read as finding a *control policy* $\Theta^*$, which minimizes the cost; note, each control policy $\Theta$ generates a dynamical system that gives the costs $\Lambda[\mathscr{K}(\Theta)]$ and $J^\Theta(X_0; c)$ in this case. However, we mention that the parameter $\Theta$ can be the physics parameters used to design the robot body for automated fabrication or any parameter that uniquely determines the dynamics.

**Example 3.1** (Examples of $\Lambda$). Some of the examples of $\Lambda$ are:

1. $\Lambda[\mathscr{A}] = \max\{1, \rho(\mathscr{A})\}$, where $\rho(\mathscr{A})$ is the spectral radius of $\mathscr{A}$, prefers stable dynamics.

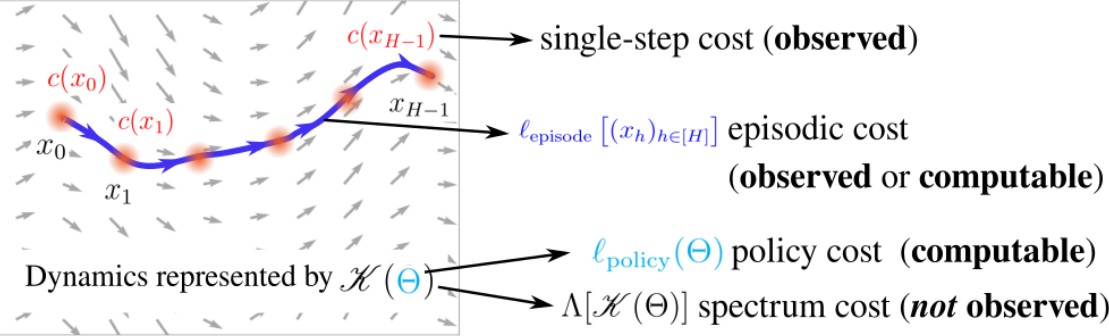

Figure 2: Comparisons of several costs for decision making problems. The Koopman spectrum cost is the cost over the global properties of the dynamical system itself which is typically unknown for learning problems, and is unobservable.

2. $\Lambda[\mathscr{A}] = \ell_{\mathscr{A}^\star}(\mathscr{A})$ can be used for imitation learning, where $\ell_{\mathscr{A}^\star}(\mathscr{A}) : \mathcal{L}(\mathcal{H}; \mathcal{H}) \to \mathbb{R}_{\geq 0}$ is a loss function measuring the gap between $\mathscr{A}$ and the given $\mathscr{A}^\star \in \mathcal{L}(\mathcal{H}; \mathcal{H})$.

3. $\Lambda[\mathscr{A}] = \sum_i |\lambda_i(\mathscr{A})|$, prefers agent behaviors described by fewer dominating modes. Here, $\{\lambda_i\}_{i \in \mathbb{Z}_{>0}}$ is the set of eigenvalues of the operator $\mathscr{A}$ (assuming that the operator has discrete spectrum).

Assuming that the Koopman operator is defined over a finite-dimensional space, an eigenvalue of the operator corresponding to each eigenfunction can be given by that of the matrix realization of the operator. In practice, one employs a finite-dimensional space even if it is not invariant under the Koopman operator; in such cases, there are several analyses that have recently been made for providing estimates of the spectra of the Koopman operator through computations over such a finite-dimensional space. Interested readers may be referred to (Ishikawa et al., 2024; Colbrook & Townsend, 2024; Colbrook et al., 2023) for example. In particular, avoiding *spectral pollution*, which refers to the phenomena where discretizations of an infinite-dimensional operator to a finite matrix create spurious eigenvalues, and approximating the continuous spectra have been actively studied with some theoretical convergence guarantees for the approximated spectra. In order to obtain an estimate of the spectral radius through computations on a matrix of finite rank, the Koopman operator may need to be compact (see the very recent work Akindji et al. (2024) for example).

**Remark 3.2** (Remarks on how the choice of $\mathcal{H}$ affects the Koopman spectrum cost)**.** As we have discussed in Remark 3.1, the mathematical properties of the Koopman operator depend on the function space $\mathcal{H}$, and information of the background dynamics implied by this operator depends on the choice of $\mathcal{H}$; however their spectral properties typically capture global characterstics of the dynamics through its linearity.

We mention that the Koopman operator over $\mathcal{H}$ may *fully represents* the dynamics in the sense that it can reproduce the dynamical system over the state space (i.e., the original space) under certain conditions (see Ishikawa et al. (2024) for example on detailed discussions); however, depending on the choice of $\mathcal{H}$, it is often the case that the Koopman operator does not uniquely reproduce the dynamics. The extreme case of such an example is the function space of single dimension spanned by a constant function, for which the Koopman operator does not carry any information about the dynamics. Even for the cases where the Koopman operator over the chosen space $\mathcal{H}$ only captures *partial information* on the dynamics, the spectrum cost is still expected to act as a regularizer of such partial spectral properties of the dynamics.

As also mentioned in Remark 3.2, the Koopman spectrum cost regularizes certain global properties of the generated dynamical system; as such, it is advantageous to employ this cost $\Lambda$ over sums of single-step costs $c(x)$ that are the objective in MDPs especially when encoding stability of the system for example. To illustrate this perspective more, we consider generating a desirable dynamical system (or trajectory) as a solution to some optimization problem.

**Regularization of global characteristic of dynamics**

Figure 3: While single-step costs (taking the current and next states as input) could be used to specify every transition, acting as a "local" cost, the Koopman spectrum cost regularizes "global" characteristics of the dynamics through specifying its spectral properties (e.g., by forcing the dynamics to have some given mode $\mathbf{m}^*$ as its top mode). The regularization incurred by the Koopman spectrum cost may not be implemented by the cumulative cost formulation in a straightforward manner. We mention it has some relations to the skill learning with motor primitives (see Section 2) in the sense that both aim at regulating the global dynamical properties.

Let $\mathcal{X} = \mathbb{R}$, $\upsilon \in (0,1]$, and let $c : \mathcal{X} \to \mathbb{R}_{\geq 0}$ be nonconstant cost function. We consider the following loss function for a random dynamical system $\mathcal{F} : \mathbb{N} \times \Omega \times \mathcal{X} \to \mathcal{X}$:

$$\ell(\mathcal{F}, x) := \mathbb{E}_\Omega \sum_{h=0}^\infty \upsilon^h c\left(\mathcal{F}(h, \omega, x_0)\right).$$

Now consider the dynamical system $\mathcal{F}(1, \omega, x) = -x$, $\forall x \in \mathcal{X}$, $\forall \omega \in \Omega$, for example. Then, it holds that, for any choice of $\upsilon$ and $c$, there exists another random dynamical system $\mathcal{G}$ satisfying that for all $x \in \mathcal{X}$,

$$\ell(\mathcal{G}, x) < \ell(\mathcal{F}, x).$$

This fact indicates that there exists a dynamical system that cannot be described by a solution to the above optimization. Note, however, that given any deterministic map $f^\star : \mathcal{X} \to \mathcal{X}$, if one employs a (different form of) cost $\mathbf{c} : \mathcal{X} \times \mathcal{X} \to \mathbb{R}_{\geq 0}$, where $\mathbf{c}(x, y)$ evaluates to 0 only if $y = f^\star(x)$ and otherwise evaluates to 1, it is straightforward to see that the dynamics $f^\star$ is the one that simultaneously optimizes $\mathbf{c}(x, y)$ for any $x \in \mathcal{X}$. In other words, this form of single-step cost uniquely identifies the (deterministic) dynamics by defining its evaluation at each state (see Figure 3 (Left)).

In contrast, when one wishes to constrain or regularize the dynamics globally in the (spatio-temporal) spectrum domain, to obtain the set of stable dynamics for example, single-step costs become powerless. Intuitively, while cumulative cost can effectively determine or regularize one-step transitions towards certain state, the Koopman spectrum cost can characterize the dynamics globally (see Figure 3 (Right)). Refer to Appendix C for more formal arguments.

Although aforementioned facts are simple, they give us some insights on the limitation of the use of (cumulative) single-step cost for characterizing dynamical systems, and imply that enforcing certain constraints on the dynamics requires the Koopman spectrum perspective. This is similar to the problems treated in the Fourier analysis; where the global (spectral) characteristics of the sequential data are better captured in the frequency domain.

Lastly, we depicted how the spectrum cost differs from other types of costs used in decision making problems in Figure 2. The figure illustrates that the spectrum cost is not incurred on a single-step or on one trajectory

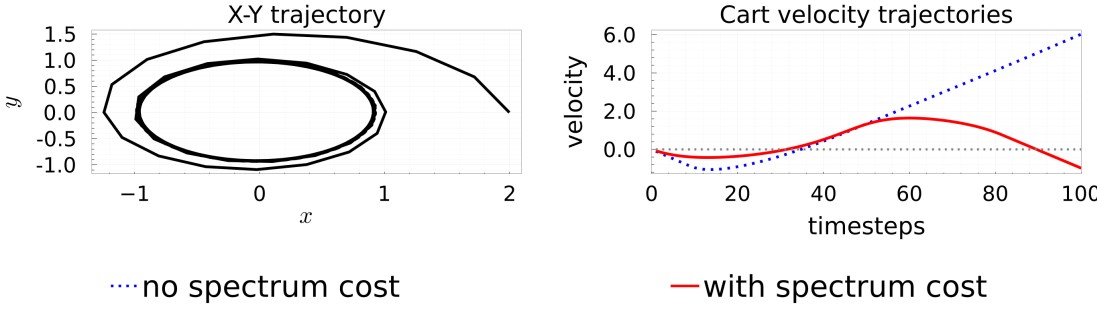

Figure 4: Left: We minimize solely for Koopman spectrum cost $\Lambda(\mathscr{A}) = \|\mathbf{m} - \mathbf{m}^\star\|_1$ to imitate the top mode of a reference spectrum to recover a desired limit cycle behavior for the single-integrator system. Right: By regularizing the spectral radius of Cartpole with a cumulative cost that favors high velocity, the cartpole performs a stable oscillation rather than moving off to infinity.

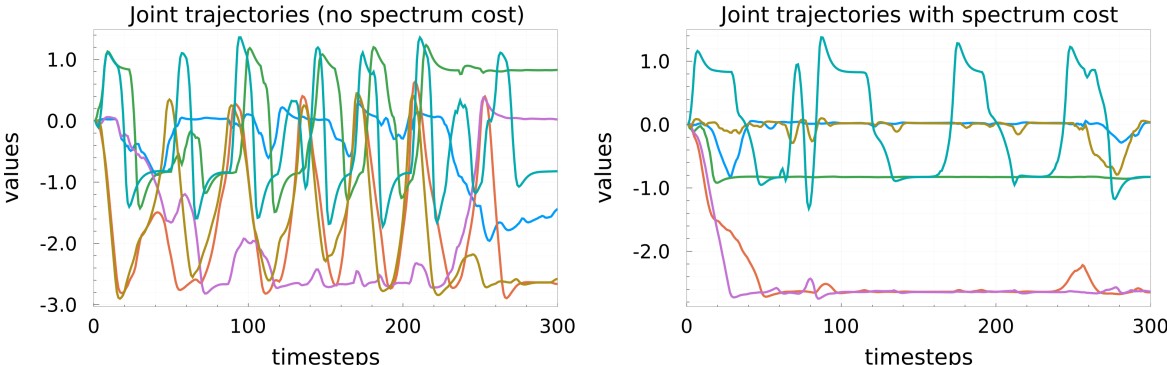

Figure 5: The joint angle trajectories generated by a combination of linear and RFF policies. Left: when only cumulative reward is maximized. Right: when both the cumulative cost and the spectrum cost $\Lambda(\mathscr{A}) = 5 \sum_{i=1}^{d_\phi} |\lambda_i(\mathscr{A})|$ are used, where the factor 5 is multiplied to balance between the spectrum cost and the cumulative cost.

(episode), but is over a (part of) the global properties of the dynamical system model (see also Remark 3.2). The dynamics typically corresponds to a policy, but a policy regularization cost used in, for example (Haarnoja et al., 2018), is computable when given the current policy while the spectrum cost is unobservable if the dynamics is unknown (and hence is not directly computable).

## 3.3 Simulated experiments

We illustrate how one can use the costs in Example 3.1. See Appendix H for detailed descriptions and results of the experiments. Throughout, we used Julia language (Bezanson et al., 2017) based robotics control package, Lyceum (Summers et al., 2020), for simulations and visualizations. Also, we use Cross-Entropy Method (CEM) based policy search (Kobilarov (2012); one of the population based policy search techniques) to optimize the policy parameter $\Theta$ to minimize the cost in (3.2). Specifically, at each iteration of CEM, we generate many parameters ($\Theta$s) to compute the loss (i.e., the sum of the Koopman spectrum cost and *negative* cumulative reward). This is achieved by fitting the transition data to the chosen feature space to estimate its (finite-dimensional realization of) Koopman operator (see Appendix H.1); here the data are generated by the simulator which we assume to have access to.

**Imitating target behaviors through the Koopman operators**   We consider the limit cycle dynamics

$$\dot{r} = r(1 - r^2), \ \dot{\theta} = 1,$$

described by the polar coordinates, and find the Koopman operator for this dynamics by sampling transitions, assuming $\mathcal{H}$ is the span of Random Fourier Features (RFFs) (Rahimi & Recht, 2007). We illustrate how KSNR is used to imitate the dynamics; in particular, by imitating the *Koopman modes*, we expect that some physically meaningful dynamical behaviors of the target system can be effectively reconstructed.

To define Koopman modes, suppose that the Koopman operator $\mathscr{A}^\star$ induced by the target dynamics has eigenvalues $\lambda_i \in \mathbb{C}$ and eigenfunctions $\xi_i : \mathcal{X} \to \mathbb{C}$ for $i \in \{1, 2, \ldots, d_\phi\}$, i.e.,

$$\mathscr{A}^\star \xi_i = \lambda_i \xi_i.$$

If the set of observables $\phi_i$s satisfies

$$\boldsymbol{\phi}_x = \sum_{i=1}^{d_\phi} \xi_i(x) \mathbf{m}_i^\star,$$

for $\mathbf{m}_i^\star \in \mathbb{C}^{d_\phi}$, where $\boldsymbol{\phi}_x := [\phi_1(x), \phi_2(x), \ldots, \phi_{d_\phi}(x)]^\top \in \mathbb{R}^{d_\phi}$, then $\mathbf{m}_i^\star$s are called the Koopman modes. The Koopman modes are closely related to the concept *isostable*; interested readers are referred to (Mauroy et al., 2013) for example.

In this simulated example, the target system is being imitated by forcing the generated dynamics to have (approximately) the same top mode (i.e., the Koopman modes corresponding to the largest absolute eigenvalue) that dominates the behavior. To this end, with $\Pi$ a space of RFF policies that define $\dot{r}$ and $\dot{\theta}$ as a single-integrator model, we solve KSNR for the spectrum cost $\Lambda(\mathscr{A}) = \|\mathbf{m} - \mathbf{m}^\star\|_1$, where $\mathbf{m} \in \mathbb{C}^{d_\phi}$ and $\mathbf{m}^\star$ are the top modes of the Koopman operator induced by the generated dynamics and of the target Koopman operator found previously, respectively.

Figure 4 (Left) plots the trajectories (of the Cartesian coordinates) generated by RFF policies that minimize this cost; it is observed that the agent successfully converged to the desired limit cycle of radius one by imitating the dominant mode of the target spectrum.

**Generating stable loops (Cartpole)**   We consider Cartpole environment (where the rail length is extended from the original model). The single-step reward (negative cost) is $10^{-3}|v|$ where $v$ is the velocity of the cart, plus the penalty $-1$ when the pole falls down (i.e., directing downward). This single-step reward encourages the cartpole to maximize its velocity while preferring not to let the pole fall down. The additional spectrum cost considered in this experiment is $\Lambda(\mathscr{A}) = 10^4 \max(1, \rho(\mathscr{A}))$, which highly penalizes spectral radius larger than one; it therefore regularizes the dynamics to be stable, preventing the velocity from ever increasing its magnitude.

Figure 4 (Right) plots the cart velocity trajectories generated by RFF policies that (approximately) solve KSNR with/without the spectrum cost. It is observed that spectral regularization led to a back-and-forth motion while the non-regularized policy preferred accelerating to one direction to solely maximize velocity. When the spectrum cost was used, the cumulative rewards were 0.072 and the spectral radius was 0.990, while they were 0.212 and 1.003 when the spectrum cost was not used; limiting the spectral radius prevents the ever increasing change in position.

We mention that this experiment is not intended to force the dynamics to have this oscillation, but rather to let the dynamics be stable in the sense that the Koopman operator over a chosen space has spectral radius that is less than or equal to one (see Figure 3 to review the difference of concepts and roles of cumulative cost and Koopman spectrum cost). See more experimental results in Appendix I.

**Remark 3.3** (Remarks on the stability regularization). As mentioned in Section 2, this stability regularization is not meant to learn the stable Koopman operator but to find an RDS that balances the cumulative cost and the spectrum cost that enforces stability; in other words, the task is to find a dynamical system that minimizes the cumulative cost under the hard stability constraint encoded by the spectrum cost. Here,

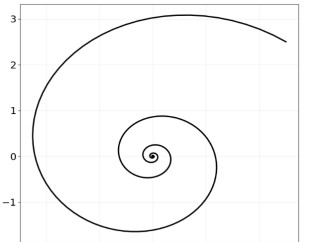 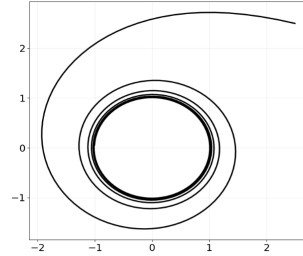 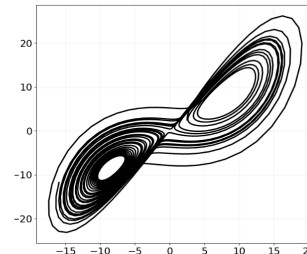

Figure 6: Examples of attractor dynamics that are stable; from the left, they are a fixed-point attractor, limit cycle attractor, and strange attractor. Those should be included in the set of stable dynamics.

we emphasize that the set of stable dynamics should include a variety of dynamics ranging from a fixed-point attractor to a strange attractor as portrayed in Figure 6. On the other hand, constraining dynamics by a single control Lyapunov function (cf. Freeman & Primbs (1996)) will form a strictly smaller set of stable dynamics (see the discussions in Section 3.2 as well). Also, in this simulation example, the stability does not necessarily mean the behaviors converging to the zero state as long as the velocity does not increase indefinitely, and the preference of keeping the pole straight up is encoded by the cumulative cost term instead.

**Generating smooth movements (Walker)** We use the Walker2d environment and compare movements with/without the spectrum cost. The single-step reward (negative cost) is given by $v - 0.001\|a\|_{\mathbb{R}^6}^2$, where $v$ is the velocity and $a$ is the action vector of dimension 6. The spectrum cost is given by $\Lambda(\mathscr{A}) = 5\sum_{i=1}^{d_\phi} |\lambda_i(\mathscr{A})|$, where the factor 5 is multiplied to balance between the spectrum and cumulative cost. The single-step reward encourages the walker to move to the right as fast as possible with a small penalty incurred on the action input, and the spectrum cost regularizes the dynamics to have fewer dominating modes, which intuitively leads to *smoother* motions.

Figure 5 plots typical joint angles along a trajectory generated by a combination of linear and RFF policies that (approximately) solves KSNR with/without the spectrum cost. It is observed that the spectrum cost led to simpler (i.e., some joint positions converge) and smoother dynamics while doing the task sufficiently well. With the spectrum cost, the cumulative rewards and the spectrum costs averaged across four random seeds (plus-minus standard deviation) were $584 \pm 112$ and $196 \pm 8.13$. Without the spectrum cost, they were $698 \pm 231$ and $310 \pm 38.6$. We observe that, as expected, the spectrum cost is lower for KSNR while the classical cumulative reward is higher for the behavior generated by the optimization without spectrum cost. Again, we emphasize that we are *not* competing against the MDP counterparts in terms of the cumulative reward, but rather showing an effect of additional spectrum cost. Please also refer to Appendix H and I for detailed results.

## 4 Theoretical algorithm of online KSNR and its insights on the complexity

In this section, we present a (theoretical) learning algorithm for online KSNR. Although the KSNR itself is a general regulator framework, we need some structural assumptions to the problem in order to devise a sample efficient (if not computation efficient) algorithm. Nevertheless, despite the unobservability of the spectrum cost, KSNR admits a sample efficient model-based algorithm through operator theoretic arguments under those assumptions. Here, "model-based" simply means we are not directly learning the spectrum cost itself but the Koopman operator model. We believe the algorithm and its theoretical analysis clarify the intrinsic difficulty of considering the spectrum cost, and pave the way towards future research.

The learning goal is to find a parameter $\Theta^{\star t}$ that satisfies (3.2) at each episode $t \in [T]$. We employ episodic learning, and let $\Theta^t$ be a parameter employed at the $t$-th episode. Adversary chooses $X_0^t := \{(x_{0,0}^t, H_0^t), (x_{0,1}^t, H_1^t), \ldots, (x_{0,N^t-1}^t, H_{N^t-1}^t)\} \subset \mathcal{X} \times \mathbb{Z}_{>0}$, where $N^t \in \mathbb{Z}_{>0}$, and the cost function $c^t$ at the beginning of each episode. Let $c_{h,n}^t := c^t(x_{h,n}^t)$. $\omega \in \Omega_{\Theta^t}$ is chosen according to $P_{\Theta^t}$.

**Algorithm evaluation.** In this work, we employ the following performance metric, namely, the cumulative regret:

$$\text{REGRET}_T := \sum_{t=0}^{T-1} \left( \Lambda[\mathscr{K}(\Theta^t)] + \sum_{n=0}^{N^t-1} \sum_{h=0}^{H_n^t-1} c_{h,n}^t \right) - \sum_{t=0}^{T-1} \min_{\Theta \in \Pi} \left( \Lambda[\mathscr{K}(\Theta)] + J^{\Theta}(X_0^t; c^t) \right).$$

Note the minimum on the second term on the right hand side is taken at every episode. Below, we present model assumptions and an algorithm with a regret bound.

## 4.1 Models and algorithms

We make the following modeling assumptions.

**Assumption 1.** *Let $\mathscr{K}(\Theta)$ be the Koopman operator corresponding to a parameter $\Theta$. Then, assume that there exists a finite-dimensional subspace $\mathcal{H}_0$ on $\mathcal{X}$ over $\mathbb{R}$ and its basis $\phi_1, \ldots, \phi_{d_\phi}$ such that the random dynamical system (3.1) satisfies the following:*

$$\forall \Theta \in \Pi, \ \forall x \in \mathcal{X}: \ \phi_i(\mathcal{F}^{\Theta}(1, \omega, x)) = [\mathscr{K}(\Theta)\phi_i](x) + \epsilon_i(\omega),$$

*where the additive noise $\epsilon_i(\omega) \sim \mathcal{N}(0, \sigma^2)$ is assumed to be independent across timesteps, parameters $\Theta$, and indices $i$.*

**Remark 4.1** (On Assumption 1). Although the added noise term is expected to deal with stochasticity and misspecification to some extent in practice, Assumption 1 is strong to ask for; in fact, claiming existence of the Koopman operator over a useful RKHS (e.g., with Gaussian kernel) is not trivial for most of the practical problems. Studying *misspecified case* with small error margin is an important future direction of research; however, as our regulator problem is novel, we believe this work guides the future attempts of further algorithmic research.

**Assumption 2** (Function-valued RKHS (see Appendix A for details)). *$\mathscr{K}(\cdot)\phi$ for $\phi \in \mathcal{H}$ is assumed to live in a known function valued RKHS with the operator-valued reproducing kernel $K(\cdot, \cdot) : \Pi \times \Pi \to \mathcal{L}(\mathcal{H}; \mathcal{H})$, or equivalently, there exists a known map $\Psi : \Pi \to \mathcal{L}(\mathcal{H}; \mathcal{H}')$ for a specific Hilbert space $\mathcal{H}'$ satisfying for any $\phi \in \mathcal{H}$, there exists $\psi \in \mathcal{H}'$ such that*

$$\mathscr{K}(\cdot)\phi = \Psi(\cdot)^{\dagger}\psi. \tag{4.1}$$

**Remark 4.2** (On Assumption 2). The assumption intuitively states that the *structure* on how the closeness of parameters $\Theta$s relates to that of the resulting Koopman operators is known. One can always consider an expressive function-valued RKHS in practice, but it may lead to increased (effective) dimensions that will require more samples to learn.

When Assumption 2 holds, we obtain the following claim which is critical for our learning framework.

**Lemma 4.3.** *Suppose Assumption 2 holds. Then, there exists a linear operator $M^{\star} : \mathcal{H} \to \mathcal{H}'$ such that*

$$\mathscr{K}(\Theta) = \Psi(\Theta)^{\dagger} \circ M^{\star}.$$

In the reminder of this paper, we work on the invariant subspace $\mathcal{H}_0$ in Assumption 1 and thus we regard $\mathcal{H} = \mathcal{H}_0 \cong \mathbb{R}^{d_\phi}$, $\mathcal{L}(\mathcal{H}; \mathcal{H}) \cong \mathbb{R}^{d_\phi \times d_\phi}$, and, by abuse of notations, we view $\mathscr{K}(\Theta)$ as the realization of the operator over $\mathbb{R}^{d_\phi}$, i.e.,

$$\phi_{\mathcal{F}^{\Theta}(1,\omega,x)} = \mathscr{K}(\Theta)\phi_x + \epsilon(\omega) = [\Psi(\Theta)^{\dagger} \circ M^{\star}]\phi_x + \epsilon(\omega),$$

where $\phi_x := [\phi_1(x), \phi_2(x), \ldots, \phi_{d_\phi}(x)]^{\top} \in \mathbb{R}^{d_\phi}$, and $\epsilon(\omega) := [\epsilon_1(\omega), \epsilon_2(\omega), \ldots, \epsilon_{d_\phi}(\omega)]^{\top} \in \mathbb{R}^{d_\phi}$.

Finally, we assume the following.

**Assumption 3** (Realizability of costs). *For all $t$, the single-step cost $c^t$ is known and satisfies $c^t(x) = w^t(\phi_x)$ for some known map $w^t : \mathbb{R}^{d_\phi} \to \mathbb{R}_{\geq 0}$.*

---

**Algorithm 1** Koopman-Spectrum LC$^3$ (KS-LC$^3$)

---

**Require:** Parameter set $\Pi$; regularizer $\lambda$
 1: Initialize $\text{BALL}_M^0$ to be a set containing $M^\star$.
 2: **for** $t = 0 \ldots T - 1$ **do**
 3:  Adversary chooses $X_0^t$.
 4:  $\Theta^t, \hat{M}_t = \arg\min_{\Theta \in \Pi,\ M \in \text{BALL}_M^t} \Lambda[\Psi(\Theta)^\dagger \circ M] + J^\Theta(X_0^t; M; c^t)$
 5:  Under the dynamics $\mathcal{F}^{\Theta^t}$, sample transition data $\tau^t := \{\tau_n^t\}_{n=0}^{N^t-1}$, where $\tau_n^t := \{x_{h,n}^t\}_{h=0}^{H_n^t}$
 6:  Update $\text{BALL}_M^{t+1}$.
 7: **end for**

---

**Remark 4.4** (On Assumption 3). As mentioned in Remark 3.2, the space $\mathcal{H}_0$ over which the Koopman operator is acting should be properly chosen so that the Koopman operator exists and that its spectrum cost has desirable regularization effect over the dynamics. At the same time, Assumption 3 requires that $\mathcal{H}_0$ is sufficiently *expressive* in the sense that it can capture the immediate cost $c^t$.

For later use, we define, for all $t \in [T]$, $n \in [N^t]$, and $h \in [H_n^t]$; $\mathscr{A}_{h,n}^t \in \mathcal{L}\left(\mathcal{L}(\mathcal{H}; \mathcal{H}'); \mathcal{H}\right)$ and $\mathscr{B}^t \in \mathcal{L}\left(\mathcal{L}(\mathcal{H}; \mathcal{H}'); \mathcal{L}(\mathcal{H}; \mathcal{H})\right)$ by

$$\mathscr{A}_{h,n}^t(M) = \left[\Psi(\Theta^t)^\dagger \circ M\right]\left(\boldsymbol{\phi}_{x_{h,n}^t}\right), \qquad \mathscr{B}^t(M) = \Psi(\Theta^t)^\dagger \circ M.$$

**Remark 4.5** (Hilbert-Schmidt operators). Both $\mathscr{A}_{h,n}^t$ and $\mathscr{B}^t$ are Hilbert-Schmidt operators because the ranges $\mathcal{H}$ and $\mathcal{L}(\mathcal{H}; \mathcal{H})$ are of finite dimension. Note, in case $\mathcal{H}'$ is finite, we obtain

$$\boldsymbol{\phi}_{\mathcal{F}^\Theta(1,\omega,x)} = \Psi(\Theta)^\dagger M^\star \boldsymbol{\phi}_x + \epsilon(\omega) = (\boldsymbol{\phi}_x^\dagger \otimes \Psi(\Theta)^\dagger)\text{vec}(M^\star) + \epsilon(\omega), \tag{4.2}$$

where vec is the vectorization of matrix.

With these preparations in mind, our proposed information theoretic algorithm, which is an extension of LC$^3$ to KSNR problem (estimating the true operator $M^\star$) is summarized in Algorithm 1.[1] This algorithm assumes the following oracle.

**Definition 4.1** (Optimal parameter oracle). Define the oracle, `OptDynamics`, that computes the minimization problem (3.2) for any $\mathscr{K}$, $X_0$, $\Lambda$ and $c^t$ satisfying Assumption 3.

## 4.2 Information theoretic regret bounds

Here, we present the regret bound analysis. To this end, we make the following assumptions.

**Assumption 4.** *The operator $\Lambda$ satisfies the following (modified) Hölder condition:*

$$\exists L \in \mathbb{R}_{\geq 0},\ \exists \alpha \in (0, 1],\ \forall \mathscr{A} \in \mathcal{L}\left(\mathcal{H}, \mathcal{H}\right),\ \forall \Theta \in \Pi,$$

$$|\Lambda[\mathscr{A}] - \Lambda[\mathscr{K}(\Theta)]| \leq L \cdot \max\left\{\|\mathscr{A} - \mathscr{K}(\Theta)\|^2, \|\mathscr{A} - \mathscr{K}(\Theta)\|^\alpha\right\}.$$

*Further, we assume there exists a constant $\Lambda_{\max} \geq 0$ such that, for any $\Theta \in \Pi$ and for any $M \in \text{BALL}_M^0$,*

$$\left|\Lambda[\Psi(\Theta)^\dagger \circ M]\right| \leq \Lambda_{\max}.$$

**Remark 4.6** (On Assumption 4). This assumption does not preclude practical examples such as matrix norms (because of the triangle inequality) and the spectral radius as described below. However, we note that the spectrum cost in Section 3.3 used for top mode imitation, namely, $\Lambda(\mathscr{A}) = \|\mathbf{m}_1 - \mathbf{m}_1^\star\|_1$, does not satisfy this (modified) Hölder condition in general; therefore this cost might not be used for KS-LC$^3$.

For example, for spectral radius $\rho$, the following proposition holds using the result from (Song, 2002, Corollary 2.3).

---

[1] See Appendix B for the definitions of the values in this algorithm.

**Proposition 4.7.** *Assume there exists a constant $\Lambda_{\max} \geq 0$ such that, for any $\Theta \in \Pi$ and for any $M \in BALL_M^0$, $\rho(\Psi(\Theta)^\dagger \circ M) \leq \Lambda_{\max}$. Let the Jordan condition number of $\mathscr{K}(\Theta)$ be the following:*

$$\kappa := \sup_{\Theta \in \Pi} \inf_{\mathcal{Q}(\Theta)} \left\{ \|\mathcal{Q}(\Theta)\| \|\mathcal{Q}(\Theta)^{-1}\| : \mathcal{Q}(\Theta)^{-1}\mathscr{K}(\Theta)\mathcal{Q}(\Theta) = \mathscr{J}(\Theta) \right\},$$

*where $\mathscr{J}(\Theta)$ is a Jordan canonical form of $\mathscr{K}(\Theta)$ transformed by a nonsingular matrix $\mathcal{Q}(\Theta)$. Also, let $m$ be the order of the largest Jordan block. Then, if $\kappa < \infty$, the cost $\Lambda(\mathscr{A}) = \rho(\mathscr{A})$ satisfies the Assumption 4 for*

$$L := (1 + \kappa)d_\phi^2(1 + \sqrt{d_\phi - 1}), \quad \alpha = \frac{1}{m}.$$

Note when $\mathscr{K}(\Theta)$ is diagonalizable for all $\Theta$, $\alpha = 1$.

**Assumption 5.** *For every $t$, adversary chooses $N^t$ trajectories such that $\{\phi_{x_{0,n}^t}\}$ satisfies that the smallest eigenvalue of $\sum_{n=0}^{N^t-1} \phi_{x_{0,n}^t} \phi_{x_{0,n}^t}^\dagger$ is bounded below by some constant $C > 0$. Also, there exists a constant $H \in \mathbb{Z}_{>0}$ such that for every $t$, $\sum_{n=0}^{N^t-1} H_n^t \leq H$.*

**Remark 4.8** (On Assumption 5). In practice, user may wait to end an episode until a sufficient number of trajectories is collected in order for the assumption to hold. Intuitively, this condition ensures that fitting the transition data over the feature space is not ill-posed. Note, this assumption does not preclude the necessity of exploration: If the sole purpose is just to fit the data for single parameter $\Theta$, we may not need a well-designed exploration strategy in this case; however, because the smallest eigenvalue of $\sum_{n=0}^{N^t-1} \sum_{h=0}^{H_n^t-1} \mathscr{A}_{h,n}^t {}^\dagger \mathscr{A}_{h,n}^t$ is in general not bounded below by a positive constant, we still need careful design of exploration.

We mention that this assumption plays a role in our regret analysis to simultaneously manage the cumulative cost and the spectrum cost through the use of common confidence balls; note the former cares about each single-step transition while the latter deals with the global dynamical properties represented as the Koopman operator realized over a given space.

The direct use of this assumption is highlighted by Lemma F.2 in Appendix F (derived from our *positive operator norm bounding lemma*; see Lemma F.1) which is used to basically bound the norm of difference between the true Koopman operator and the estimated one at each episode by the multiple of *radius* of the confidence ball.

We hope that this assumption can be relaxed by assuming the bounds on the norms of $\phi_{x_{0,n}}$ and $\mathscr{K}(\Theta)$ and by using matrix Bernstein inequality under additive Gaussian noise assumption.

Lastly, the following assumption is the modified version of the one made in (Kakade et al., 2020).

**Assumption 6.** *[Modified version of Assumption 2 in (Kakade et al., 2020)] Assume there exists a constant $V_{\max} > 0$ such that, for every $t$,*

$$\sup_{\Theta \in \Pi} \sum_{n=0}^{N^t-1} \mathbb{E}_{\Omega_\Theta} \left[ \left( \sum_{h=0}^{H_n^t-1} c^t(x_{h,n}^t) \right)^2 \;\middle|\; \Theta, x_{0,n}^t \right] \leq V_{\max}.$$

**Remark 4.9** (On Assumption 6). This is a slight modification of Assumption 2 in (Kakade et al., 2020) to adjust to our problem settings; this assumption *does not* state that the cost function is bounded over the state space but the second moment of *realized* cumulative cost is bounded, and the complexity depends on this bound.

**Theorem 4.10.** *Suppose Assumption 1 to 6 hold. Set $\lambda = \frac{\sigma^2}{\|M^\star\|^2}$. Then, there exists an absolute constant $C_1$ such that, for all $T$, KS-$LC^3$(Algorithm 1) using `OptDynamics` enjoys the following regret bound:*

$$\mathbb{E}_{\text{KS}-\text{LC}^3}[REGRET_T] \leq C_1(\tilde{d}_{T,1} + \tilde{d}_{T,2})T^{1-\frac{\alpha}{2}},$$

*where*

$$\tilde{d}_{T,1}^2 := (1 + \gamma_{T,\mathscr{B}}) \left[ L^2 (1 + C^{-1})^2 \tilde{\beta}_{2,T} + (L^2 + \Lambda_{\max} L)(1 + C^{-1}) \tilde{\beta}_{1,T} + \Lambda_{\max}^2 + L^2 \right],$$
$$\tilde{d}_{T,2}^2 := \gamma_{T,\mathscr{A}} H V_{\max} \left( \gamma_{T,\mathscr{A}} + d_\phi + \log(T) + H \right),$$
$$\tilde{\beta}_{1,T} := \sigma^2 (d_\phi + \log(T) + \gamma_{T,\mathscr{A}}),$$
$$\tilde{\beta}_{2,T} := \sigma^4 ((d_\phi + \log(T) + \gamma_{T,\mathscr{A}})^2 + \gamma_{2,T,\mathscr{A}}).$$

*Here, $\gamma_{T,\mathscr{A}}$, $\gamma_{2,T,\mathscr{A}}$, and $\gamma_{T,\mathscr{B}}$ are the expected maximum information gains that scale (poly-)logarithmically with $T$ under practical settings (see Appendix B for details).*

**Remark 4.11** (On the order). We note that, when $\alpha = 1$, we obtain the order of $\tilde{O}(\sqrt{T})$.

**Remark 4.12** (On the adversary). In our setting, the adversary only chooses the initial states, their time horizons, and the immediate cost function. The trajectories themselves are generated by the learner's parameter $\Theta$. Although the regret bound given above is valid if the assumptions hold regardless of the adversary, a caveat here is the bound $H$ and $V_{\max}$ appearing in the regret bound can potentially depend on the choice of the adversary.

The proof techniques include our *positive operator norm bounding lemma* (see Lemma F.1 in Appendix F), which is another crucial operator theoretic lemma in this work in addition to Lemma 4.3.

### 4.3 Relations to the kernelized nonlinear regulator and to eigenstructure assignments

First, we mention that our theoretical results apply some of the techniques developed in the work of KNR; and in fact, additional theoretical arguments that are necessitated for KSNR framework highlight the essential difference of our framework from the MDP counterpart.

As mentioned in Remark 4.1, for a given dynamics described by the system model studied in the KNR (i.e., the transition map from the current state and control to the next state is in a known RKHS), the existence of a (finite-dimensional) space $\mathcal{H}_0$ in Assumption 1 that can represent the given cumulative cost is in general not guaranteed. As such, the system model considered in this section is no more general than that of the KNR (although our spectrum cost formulation is indeed a generalization of that of the KNR). Now, we consider the finite dimensional description (see (4.2)) for simplicity. The equation (4.2) can be rewritten by

$$\boldsymbol{\phi}_{\mathcal{F}\Theta(1,\omega,x)} = \left( I \otimes \mathrm{vec}(M^\star)^\top \right) \mathrm{vec} \left[ \left( \boldsymbol{\phi}_x^\dagger \otimes \Psi(\Theta)^\dagger \right)^\top \right] + \epsilon(\omega),$$

and is a special case of the system model of the KNR.

We see that the model associated with our spectrum cost formulation reduces to the eigenstructure assignment problem for the linear systems described by

$$x_{h+1} = A x_h + B u_h + \epsilon, \quad A \in \mathbb{R}^{d_\mathcal{X} \times d_\mathcal{X}}, \; B \in \mathbb{R}^{d_\mathcal{X} \times d_\mathcal{U}}, \; \epsilon \sim \mathcal{N}(0, \sigma^2 I),$$

where $u \in d_\mathcal{U}$ is a control input. In particular, considering the feedback policy in the form of $u_h = K x_h$ where $K \in \mathbb{R}^{d_\mathcal{U} \times d_\mathcal{X}}$ (or $\Theta = K$ in this case), the system becomes $x_{h+1} = (A + BK) x_h + \epsilon$. By letting $\mathcal{H}_0$ be $\mathbb{R}^{d_\mathcal{X}}$ where the canonical basis is taken and by properly designing $\Psi(\Theta)$ (see Appendix G), our system model reduces to the linear case. The spectrum cost may be designed not only to constrain the eigenstructure but also to balance with the cumulative single-step cost in our framework as well.

### 4.4 Simulated experiments

**Linear case: as eigenstructure assignment problem** First, to demonstrate the correctness of our theoretical algorithm, we consider a linear system case. In particular, the following simple system is considered:

$$x_{h+1} = x_h + \Delta t \cdot u_h,$$

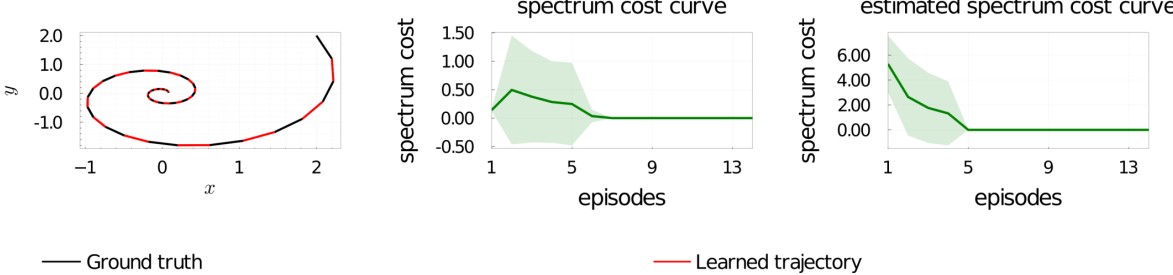

Figure 7: Linear system imitation (as eigenstructure assignment) by KS-LC$^3$ using the spectrum cost $\Lambda(\mathscr{A}) = \|(I + 0.05K^*) - (A + BK)\|^2_{HS}$ (where $A$ and $B$ are extracted from $\mathscr{A}$). Left: ground-truth linear system position trajectory and a trajectory generated by the learned model with KS-LC$^3$. We see both overlap exactly, indicating that the algorithm learned the feedback matrix properly. Middle: The spectrum cost curve evaluated approximately using the current trajectory data. Right: The estimated spectrum cost curve obtained by using the current estimate $\hat{M}$.

where $\Delta t = 0.05$. The target linear system is given by

$$x_{h+1} = (I + 0.05K^*)\, x_h,$$

where

$$K^* := \begin{bmatrix} -2 & 4 \\ -6 & -2 \end{bmatrix}.$$

The spectrum cost given by $\|(I+0.05K^*)-(A+BK)\|^2_{\mathrm{HS}}$ enforces that the parameter $\Theta$ (which is a feedback matrix $K$ in this case) generates the dynamical system that follows this target linear dynamics; where $A$ and $B$ are matrices that are part of the Koopman operator to learn (see Section 4.3). Figure 7 plots the result showing that the trajectories generated by the ground-truth target linear system and by the learned model match almost exactly, and that the spectrum cost decreases along episodes. The spectrum cost curve evaluated approximately using the current trajectory data is given in Figure 7 (Middle). Note the curve is averaged across four random seeds and across episode window of size four. Additionally, the estimated spectrum cost curve obtained by using the current estimate $\hat{M}$ is plotted in Figure 8 (Right).

**Cartpole problem with reduced policy space** We again consider a Cartpole environment for KS-LC$^3$. To reduce the policy search space, we compose three pre-trained policies to balance the pole while 1) moving the cart position $p$ to $-0.3$ and then to $0.3$, 2) moving the cart with velocity $v = -1.5$, 3) moving cart with velocity $v = 1.5$. We also extract the Koopman operator $\mathscr{A}^\star$ for the first policy. Then, we let $\Pi$ to be the space of linear combinations of the three policies, reducing the dimension of $\Theta$. We let the first four features $\phi_1(x)$ to $\phi_4(x)$ be $x_1$ to $x_4$, where $x = [x_1, \ldots, x_4]^\top$ is the state, and the rest be RFFs.

The single-step cost is given by $10^{-4}(|v| - 1.5)^2$ plus the large penalty when the pole falls down (i.e., directing downward), and the spectrum cost is $\Lambda(\mathscr{A}) = \|\mathscr{A}[1:4,:] - \mathscr{A}^\star[1:4,:]\|^2_{HS} + 0.01 \sum_{i=1}^{d_\phi} |\lambda_i(\mathscr{A})|$, where $\mathscr{A}[1:4,:] \in \mathbb{R}^{4 \times d_\phi}$ is the first four rows of $\mathscr{A}$; this imitates the first policy while also regularizing the spectrum.

When CEM is used to (approximately) solve KSNR, the resulting cart position trajectories are given by Figure 8 (Left). With the spectrum cost, the cumulative costs and the spectrum costs averaged across four random seeds were $0.706 \pm 0.006$ and $2.706 \pm 0.035$. Without the spectrum cost, they were $0.428 \pm 0.045$ and $3.383 \pm 0.604$.

We then use the Thompson sampling version of KS-LC$^3$; the resulting cart position trajectories are given by Figure 8 (Left) as well, and the spectrum cost curves are also shown. It is observed that the addition of the spectrum cost favored the behavior corresponding to the target Koopman operator to oscillate while balancing the pole, and achieved similar performance to the ground-truth model optimized with CEM.

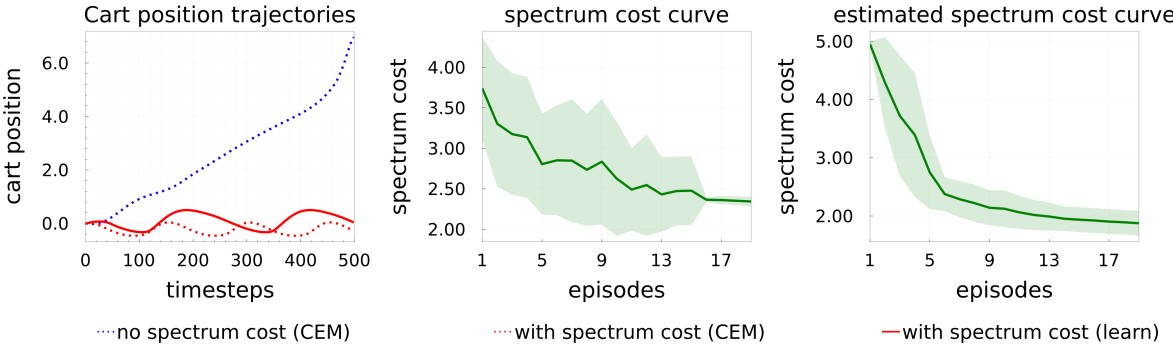

Figure 8: Cartpole imitation by KS-LC$^3$ using the spectrum cost $\Lambda(\mathscr{A}) = \|\mathscr{A}[1:4,:] - \mathscr{A}^\star[1:4,:]\|_{HS}^2 + 0.01 \sum_{i=1}^{d_\phi} |\lambda_i(\mathscr{A})|$. Left: Cart position trajectories with/without the spectrum cost with CEM on a known model versus the learned model with KS-LC$^3$. Middle: The spectrum cost curve evaluated approximately using the current trajectory data. Right: The estimated spectrum cost curve obtained by using the current estimate $\hat{M}$.

## 5 Discussion and conclusion

This work proposed a novel paradigm of regulating (controlling) dynamical systems, which we refer to as Koopman Spectrum Nonlinear Regulator, and presented an information theoretic algorithm that achieves a sublinear regret bound under model assumptions. We showed that behaviors such as limit cycles of interest, stable motion, and smooth movements are effectively realized within our framework, which is an effective generalization of classical eigenstructure (pole) assignments. In terms of learning algorithms, we stress that there is a significant potential of future work for inventing sophisticated and scalable methods that elicit desired and interesting behaviors from dynamical systems. Our motivation of this work stems from the fact that some preferences over dynamical systems cannot be encoded by cumulative single-step cost based control/learning algorithms; we believe that this work enables a broader representation of dynamical properties that can enable more intelligent and robust agent behaviors.

## 6 Limitations

First, when the dynamical systems of interests do not meet certain conditions, Koopman operators might not be well defined over functional spaces that can be practically managed (e.g. RKHSs). Studying such conditions is an on-going research topic, and we will need additional studies of misspecified cases where only an approximately invariant Koopman space is employed with errors.

Second, to solve KSNR, one needs heuristic approximations when the (policy) space $\Pi$ or the state space $\mathcal{X}$ is continuous. Even when the dynamical system model is known, getting better approximation results would require certain amount of computations, and additional studies on the relation between the amount of computations (or samples) and approximation errors would be required. Also, studying a better heuristic algorithm that is well suited to our problem to robustly scale the methodology to more complicated domains is indeed an interesting direction of research.

Third, KS-LC$^3$ requires several assumptions for tractable regret analysis. It is again a somewhat strong to assume that one can find a Koopman invariant space $\mathcal{H}$. Further, additive Gaussian noise assumption may not be satisfied exactly in some practical applications, and robustness against the violations of the assumptions is an important future work. However, we stress that we believe the analysis of provably correct methods deepens our understandings of the problem. Eventually, we hope our framework will match the maturity of the MDP counterpart, for example by studying gradient-based algorithms that scale better to the more complicated domains and their theoretical analysis with a sample complexity guarantee.

Lastly, we note that our empirical experiment results are only conducted on simulators; to apply to real robotics problems, we need additional studies, such as computation-accuracy trade-off, safety/robustness issues, and simulation-to-real, if necessary. Also, balancing between cumulative cost and the Koopman spectrum cost is necessary to avoid unexpected negative outcomes.

## 7 Broader impact statement

The work, along with other robotics/control literature, encourages the further automation of robots; which would lead to loss of jobs that are currently existent, or may lead to developments of harmful military robots. Further, when our proposed framework requires additional (computational or other) resources, it would benefit entities with larger amount of such resources. Also, there is always a risk of robots misbehaving under partially unknown environments or under adversarial attacks.

However, overall, we believe our work alone does not immediately carry a significant risk of harm. Moreover, we stress that the use of the Koopman spectrum cost has no implications on "good" and "bad" about any particular human behaviors.

## Acknowledgments

We thank the constructive comments by anonymous reviewers for improving this work. Motoya Ohnishi thanks Colin Summers for instructions on Lyceum. Kendall Lowrey and Motoya Ohnishi thank Emanuel Todorov for valuable discussions and Roboti LLC for computational supports. This work of Motoya Ohnishi, Isao Ishikawa, Masahiro Ikeda, and Yoshinobu Kawahara was supported by JST CREST Grant Number JPMJCR1913, including AIP challenge program, Japan. Also, Motoya Ohnishi is supported in part by Funai Overseas Fellowship. Sham Kakade acknowledges funding from the ONR award N00014-18-1-2247.

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

## A    Function-valued RKHS

Function-valued reproducing kernel Hilbert spaces (RKHSs) are defined below.

**Definition A.1** ([Kadri et al. (2016)](#)). *A Hilbert space $(\mathcal{H}_K, \langle \cdot, \cdot \rangle_{\mathcal{H}_K})$ of functions from $\Pi$ to a Hilbert space $(\mathcal{H}, \langle \cdot, \cdot \rangle_{\mathcal{H}})$ is called a reproducing kernel Hilbert space if there is a nonnegative $\mathcal{L}(\mathcal{H}; \mathcal{H})$-valued kernel $K$ on $\Pi \times \Pi$ such that:*

1. *$\Theta \mapsto K(\Theta', \Theta)\phi$ belongs to $\mathcal{H}_K$ for all $\Theta' \in \Pi$ and $\phi \in \mathcal{H}$,*

2. *for every $\mathscr{G} \in \mathcal{H}_K$, $\Theta \in \Pi$ and $\phi \in \mathcal{H}$, $\langle \mathscr{G}, K(\Theta, \cdot)\phi \rangle_{\mathcal{H}_K} = \langle \mathscr{G}(\Theta), \phi \rangle_{\mathcal{H}}$.*

For function-valued RKHSs, the following proposition holds.

**Proposition A.1** (Feature map ([Brault et al., 2016](#))). *Let $\mathcal{H}'$ be a Hilbert space and $\Psi : \Pi \to \mathcal{L}(\mathcal{H}; \mathcal{H}')$. Then the operator $W : \mathcal{H}' \to \mathcal{H}^{\Pi}$ defined by $[W\psi](\Theta) := \Psi(\Theta)^{\dagger}\psi$, $\forall \psi \in \mathcal{H}'$ and $\forall \Theta \in \Pi$, is a partial isometry from $\mathcal{H}'$ onto the reproducing kernel Hilbert space $\mathcal{H}_K$ with a reproducing kernel $K(\Theta_2, \Theta_1) = \Psi(\Theta_2)^{\dagger}\Psi(\Theta_1)$, $\forall \Theta_1, \Theta_2 \in \Pi$.*

**Remark A.2** (Decomposable kernel). In practice, one can use *decomposable kernel* ([Brault et al., 2016](#)); when the kernel $K$ is given by $K(\Theta_1, \Theta_2) = k(\Theta_1, \Theta_2)A$ for some scalar-valued kernel $k(\Theta_1, \Theta_2)$ and for positive semi-definite operator $A \in \mathcal{L}(\mathcal{H}, \mathcal{H})$, the kernel $K$ is called decomposable kernel. For an RKHS of a decomposable kernel $K$, (4.1) becomes

$$\mathscr{K}(\Theta)\phi = (\zeta(\Theta)^{\dagger} \otimes B)\psi,$$

where $\zeta : \Pi \to \mathcal{H}''$ is known ($\mathcal{H}''$ is some Hilbert space), and $A = BB^{\dagger}$. Further, to use RFFs, one considers a shift-invariant kernel $k(\Theta_1, \Theta_2)$.

## B    Some definitions of the values

The value $J^{\Theta}(X_0^t; M; c^t)$ in Algorithm 1 is defined by

$$J^{\Theta}(X_0^t; M; c^t) := \sum_{n=0}^{N^t-1} \mathbb{E}_{\Omega_{\Theta}} \left[ \sum_{h=0}^{H_n^t-1} c^t(x_{h,n}^t) \Big| \Theta, M, x_{0,n}^t \right],$$

where the expectation is taken over the trajectory following $\phi_{x_{h+1}} = [\Psi(\Theta)^{\dagger} \circ M]\phi_{x_h} + \epsilon(\omega)$. Also the confidence ball at $t$ is given by

$$\text{BALL}_M^t := \left\{ M \Big| \left\| (\Sigma_{\mathscr{A}}^t)^{\frac{1}{2}} \left( M - \overline{M}^t \right) \right\|^2 \leq \beta_M^t \right\} \cap \text{BALL}_M^0, \quad \Sigma_{\mathscr{A}}^t := \lambda I + \sum_{\tau=0}^{t-1} \sum_{n=0}^{N^{\tau}-1} \sum_{h=0}^{H_n^{\tau}-1} \mathscr{A}_{h,n}^{\tau}{}^{\dagger} \mathscr{A}_{h,n}^{\tau},$$

and $\Sigma_{\mathscr{A}}^0 := \lambda I$, where $\beta_M^t := 20\sigma^2 \left( d_{\phi} + \log \left( t \frac{\det(\Sigma_{\mathscr{A}}^t)}{\det(\Sigma_{\mathscr{A}}^0)} \right) \right)$ and

$$\overline{M}^t := \arg\min_M \left[ \sum_{\tau=0}^{t-1} \sum_{n=0}^{N^{\tau}-1} \sum_{h=0}^{H_n^{\tau}-1} \left\| \phi_{x_{h+1,n}^{\tau}} - \mathscr{A}_{h,n}^{\tau}(M) \right\|_{\mathbb{R}^{d_{\phi}}}^2 + \lambda \|M\|_{\text{HS}}^2 \right].$$

Similar to the work ([Kakade et al., 2020](#)), we define the expected maximum information gains as:

$$\gamma_{T,\mathscr{A}}(\lambda) := 2 \max_{\mathcal{A}} \mathbb{E}_{\mathcal{A}} \left[ \log \left( \frac{\det \left( \Sigma_{\mathscr{A}}^T \right)}{\det \left( \Sigma_{\mathscr{A}}^0 \right)} \right) \right].$$

Here, det is a properly defined functional determinant of a bounded linear operator. Further,

$$\gamma_{2,T,\mathscr{A}}(\lambda) := 2 \max_{\mathcal{A}} \mathbb{E}_{\mathcal{A}} \left[ \left( \log \left( \frac{\det \left( \Sigma_{\mathscr{A}}^T \right)}{\det \left( \Sigma_{\mathscr{A}}^0 \right)} \right) \right)^2 \right].$$

Also, we define

$$\Sigma_{\mathscr{B}}^t := \lambda I + \sum_{\tau=0}^{t-1} \mathscr{B}^{\tau\dagger} \mathscr{B}^\tau, \quad \Sigma_{\mathscr{B}}^0 := \lambda I, \quad \gamma_{T,\mathscr{B}}(\lambda) := 2 \max_{\mathcal{A}} \mathbb{E}_{\mathcal{A}} \left[ \log \left( \frac{\det \left( \Sigma_{\mathscr{B}}^T \right)}{\det \left( \Sigma_{\mathscr{B}}^0 \right)} \right) \right].$$

**Lemma B.1.** *Assume that $\Psi(\Theta) \in \mathbb{R}^{d_\Psi \times d_\phi}$ and that $\|\mathscr{B}^t\|_{\mathrm{HS}} \leq B_{\mathscr{B}}$, $\|\mathscr{A}_{h,n}^t\|_{\mathrm{HS}} \leq B_{\mathscr{A}}$ for all $t \in [T]$, $n \in [N^t]$, and $h \in [H_n^t]$ and some $B_{\mathscr{B}} \geq 0$ and $B_{\mathscr{A}} \geq 0$. Then, $\gamma_{T,\mathscr{A}}(\lambda) = O(d_\phi d_\Psi \log(1 + THB_{\mathscr{A}}^2/\lambda))$, and $\gamma_{T,\mathscr{B}}(\lambda) = O(d_\phi d_\Psi \log(1 + TB_{\mathscr{B}}^2/\lambda))$.*

*Proof.* For $\gamma_{T,\mathscr{A}}(\lambda)$, from the definition of Hilbert-Schmidt norm, we have

$$\mathrm{tr} \left( \sum_{t=0}^{T-1} \sum_{n=0}^{N^t-1} \sum_{h=0}^{H_n^t-1} \mathscr{A}_{h,n}^t{}^\dagger \mathscr{A}_{h,n}^t \right) \leq THB_{\mathscr{A}}^2,$$

and the result follows from (Kakade et al., 2020, Lemma C.5). The similar argument holds for $\gamma_{T,\mathscr{B}}(\lambda)$ too. $\square$

## C   Formal argument on the limitation of cumulative costs

In this section, we present a formal argument on the limitation of cumulative costs, mentioned in Section 3.2. To this end, we give the following proposition.

**Proposition C.1.** *Let $\mathcal{X} = \mathbb{R}$. Consider the set $\mathcal{S}_f$ of dynamics converging to some point in $\mathcal{X}$. Then, for any choice of $v \in (0,1]$, set-valued map $\mathscr{S} : \mathcal{X} \to 2^{\mathcal{X}} \setminus \emptyset$, and cost $\mathbf{c}$ of the form*

$$\mathbf{c}(x,y) = \begin{cases} 0 & (y \in \mathscr{S}(x)), \\ 1 & (\text{otherwise}), \end{cases}$$

*we have $\{\mathcal{F}\} \neq \mathcal{S}_f$. Here, $\{\mathcal{F}\}$ are given by*

$$\bigcap_{x_0 \in \mathcal{X}} \left\{ \arg\min_{\mathcal{F}:\text{r.d.s.}} \mathbb{E}_\Omega \sum_{h=0}^{\infty} v^h \mathbf{c} \left( \mathcal{F}(h,\omega,x_0), \mathcal{F}(h+1,\omega,x_0) \right) \right\}.$$

*Proof.* Assume there exist $v$, a set-valued map $\mathscr{S}$, and $\mathbf{c}$ such that the set $\{\mathcal{F}^\Theta\} = \mathcal{S}_f$. Then, because $\mathcal{S}_f$ is strictly smaller than the set of arbitrary dynamics, it must be the case that $\exists x_0 \in \mathcal{X}$, $\exists y_0 \in \mathcal{X}$, $y_0 \notin \mathscr{S}(x_0)$. However, the dynamics $f^\star$, where $f^\star(x_0) = y_0$ and $f^\star(x) = x$, $\forall x \in \mathcal{X} \setminus \{x_0\}$, is an element of $\mathcal{S}_f$. Therefore, the proposition is proved. $\square$

**Remark C.2** (Interpretation of Proposition C.1)**.** Proposition C.1 intuitively says that the set of "stable dynamics" cannot be determined by specifying every single transition. This is because, the dynamics that does not follow any pre-specified transition can always be "stable".

## D   Proof of Lemma 4.3

It is easy to see that one can define a map $M_0$ so that it satisfies $M_0(\phi) = \psi$ such that Assumption 2 holds. Define

$$M^\star := PM_0,$$

where $P$ is the orthogonal projection operator onto the sum space of the orthogonal complement of the null space of $\Psi(\Theta)^\dagger$ for all $\Theta \in \Pi$, namely,

$$\overline{\sum_{\Theta \in \Pi} \ker(\Psi(\Theta)^\dagger)^\perp}.$$

Also, define $P_\Theta$ by the orthogonal projection onto

$$\ker(\Psi(\Theta)^\dagger)^\perp.$$

Then, for any $\Theta \in \Pi$, we obtain

$$P_\Theta M_0 = \Psi(\Theta)^{\dagger^+} \mathscr{K}(\Theta),$$

where $\mathscr{A}^+$ is the pseudoinverse of the operator $\mathscr{A}$, and $P_\Theta M_0$ is linear. Let $a, b \in \mathbb{R}$ and $\phi_1, \phi_2 \in \mathcal{H}$, and define $\tilde{\psi}$ by

$$\tilde{\psi} := PM_0(a\phi_1 + b\phi_2) - aPM_0(\phi_1) - bPM_0(\phi_2).$$

Because $P_\Theta PM_0 = P_\Theta M_0$, it follows that $P_\Theta \tilde{\psi} = 0$ for all $\Theta \in \Pi$, which implies

$$\tilde{\psi} \in \bigcap_{\Theta \in \Pi} \ker P_\Theta = \bigcap_{\Theta \in \Pi} \ker(\Psi(\Theta)^\dagger).$$

On the other hand, we have

$$\tilde{\psi} \in \overline{\sum_{\Theta \in \Pi} \ker(\Psi(\Theta)^\dagger)^\perp}.$$

Therefore, it follows that $\tilde{\psi} = 0$, which proves that $M^\star$ is linear.

## E    Proof of Proposition 4.7

Fix $\Theta \in \Pi$ and suppose that the eigenvalues $\lambda_i$ ($i \in \mathcal{I} := \{1, 2, \ldots, d_\phi\}$) of $\mathscr{K}(\Theta)$ are in descending order according to their absolute values, i.e., $|\lambda_1| \geq |\lambda_2| \geq \ldots \geq |\lambda_{d_\phi}|$. Given $\mathscr{A} \in \mathcal{L}(\mathcal{H}; \mathcal{H})$, suppose also that the eigenvalues $\mu_i$ ($i \in \mathcal{I}$) of $\mathscr{A}$ are in descending order according to their absolute values. Let

$$\kappa_\Theta := \inf_{\mathcal{Q}(\Theta)} \left\{ \|\mathcal{Q}(\Theta)\| \|\mathcal{Q}(\Theta)^{-1}\| : \mathcal{Q}(\Theta)^{-1} \mathscr{K}(\Theta) \mathcal{Q}(\Theta) = \mathscr{J}(\Theta) \right\},$$

where $\mathscr{J}(\Theta)$ is a Jordan canonical form of $\mathscr{K}(\Theta)$ transformed by a nonsingular matrix $\mathcal{Q}(\Theta)$. Also, let

$$\overline{\kappa}_\Theta := \max_{m \in \{1, \ldots, d_\phi\}} \left\{ \kappa_\Theta^{\frac{1}{m}} \right\}.$$

Then, by (Song, 2002, Corollary 2.3), we have

$$||\mu_i| - |\lambda_i|| \leq |\mu_i - \lambda_i| \leq \sum_i |\mu_i - \lambda_i| \leq \sum_i |\mu_{\pi(i)} - \lambda_i|$$
$$\leq d_\phi \sqrt{d_\phi}(1 + \sqrt{d_\phi - p}) \max \left\{ \kappa_\Theta \sqrt{d_\phi} \|\mathscr{A} - \mathscr{K}(\Theta)\|, (\kappa_\Theta \sqrt{d_\phi})^{\frac{1}{m}} \|\mathscr{A} - \mathscr{K}(\Theta)\|^{\frac{1}{m}} \right\},$$

for any $i \in \mathcal{I}$ and for some permutation $\pi$, where $p \in \{1, 2, \ldots, d_\phi\}$ is the number of the Jordan block of $\mathcal{Q}(\Theta)^{-1} \mathscr{K}(\Theta) \mathcal{Q}(\Theta)$ and $m \in \{1, 2, \ldots, d_\phi\}$ is the order of the largest Jordan block. Because $\sqrt{d_\phi - p} \leq \sqrt{d_\phi - 1}$ for any $p \in \{1, 2, \ldots, d_\phi\}$, and because

$$\max_{m \in \{1, \ldots, d_\phi\}} \left[ \sqrt{d_\phi} \cdot \max\{\sqrt{d_\phi}, (\sqrt{d_\phi})^{\frac{1}{m}}\} \cdot \max \left\{ \kappa_\Theta, \kappa_\Theta^{\frac{1}{m}} \right\} \right] \leq d_\phi \overline{\kappa}_\Theta,$$

it follows that

$$|\rho(\mathscr{A}) - \rho(\mathscr{K}(\Theta))| = ||\mu_1| - |\lambda_1||$$
$$\leq \overline{\kappa}_\Theta d_\phi^2 (1 + \sqrt{d_\phi - 1}) \max \left\{ \|\mathscr{A} - \mathscr{K}(\Theta)\|, \|\mathscr{A} - \mathscr{K}(\Theta)\|^{\frac{1}{m}} \right\}.$$

Since $\overline{\kappa}_\Theta < 1 + \kappa$, simple computations complete the proof.

## F   Regret analysis

Throughout this section, suppose Assumptions 1 to 6 hold.  Note that Assumption 3 is required for `OptDynamics`. We first give the *positive operator norm bounding lemma* followed by another lemma based on it.

**Lemma F.1** (Positive operator norm bounding lemma)**.** *Let $\mathcal{H}$ be a Hilbert space and $A_i$, $B_i \in \mathcal{L}(\mathcal{H};\mathcal{H})$ ($i \in \{1,2,\dots,n\}$). Assume, for all $i \in \{1,2,\dots,n\}$, that $A_i$ is positive definite. Also, assume $B_1$ is positive definite, and for all $i \in \{2,3,\dots,n\}$, $B_i$ is positive semi-definite. Then,*

$$\left\| \left( \sum_i B_i^{\frac{1}{2}} A_i B_i^{\frac{1}{2}} \right)^{-\frac{1}{2}} \left( \sum_i B_i \right) \left( \sum_i B_i^{\frac{1}{2}} A_i B_i^{\frac{1}{2}} \right)^{-\frac{1}{2}} \right\| \le \max_i \left\| A_i^{-1} \right\|.$$

*If $A_i B_i = B_i A_i$ for all $i \in \{1,2,\dots,n\}$, then*

$$\left\| \left( \sum_i A_i B_i \right)^{-\frac{1}{2}} \left( \sum_i B_i \right) \left( \sum_i A_i B_i \right)^{-\frac{1}{2}} \right\| \le \max_i \left\| A_i^{-1} \right\|.$$

*Proof.* Let $c := \max_i \left\| A_i^{-1} \right\|$. Then, we have, for all $i$,

$$I \preceq \left\| A_i^{-1} \right\| A_i \preceq c A_i,$$

from which it follows that

$$B_i \preceq c B_i^{\frac{1}{2}} A_i B_i^{\frac{1}{2}}.$$

Therefore, we obtain

$$\sum_i B_i \preceq c \sum_i B_i^{\frac{1}{2}} A_i B_i^{\frac{1}{2}}.$$

From the assumptions, $\left( \sum_i B_i^{\frac{1}{2}} A_i B_i^{\frac{1}{2}} \right)^{-1}$ exists and

$$\left( \sum_i B_i^{\frac{1}{2}} A_i B_i^{\frac{1}{2}} \right)^{-\frac{1}{2}} \left( \sum_i B_i \right) \left( \sum_i B_i^{\frac{1}{2}} A_i B_i^{\frac{1}{2}} \right)^{-\frac{1}{2}} \preceq cI,$$

from which we obtain

$$\left\| \left( \sum_i B_i^{\frac{1}{2}} A_i B_i^{\frac{1}{2}} \right)^{-\frac{1}{2}} \left( \sum_i B_i \right) \left( \sum_i B_i^{\frac{1}{2}} A_i B_i^{\frac{1}{2}} \right)^{-\frac{1}{2}} \right\| \le c.$$

The second claim follows immediately.                                                                 $\square$

**Lemma F.2.** *Suppose Assumptions 1, 2, and 5 hold. Then, it follows that, for all $t \in [T]$,*

$$\left\| (\Sigma_{\mathscr{B}}^t)^{\frac{1}{2}} \left( M - \overline{M}^t \right) \right\|^2 \le (1 + C^{-1}) \left\| (\Sigma_{\mathscr{A}}^t)^{\frac{1}{2}} \left( M - \overline{M}^t \right) \right\|^2.$$

*Proof.* Under Assumptions 1 and 2, define $\mathscr{C}^t \in \mathcal{L}\left( \mathcal{L}(\mathcal{H};\mathcal{H}'); \mathcal{L}(\mathcal{H};\mathcal{H}') \right)$ by

$$\mathscr{C}^t(M) = M \circ \left[ \sum_{n=0}^{N^t-1} \sum_{h=0}^{H_n^t-1} \phi_{x_{h,n}^t} \phi_{x_{h,n}^t}^{\dagger} \right].$$

Also, define $\mathscr{X}^t := \sum_{n=0}^{N^t-1} \sum_{h=0}^{H_n^t-1} {\mathscr{A}_{h,n}^t}^\dagger \mathscr{A}_{h,n}^t$ and $\mathscr{Y}^t := \mathscr{B}^{t\dagger} \mathscr{B}^t$. We have $\mathscr{C}^t \mathscr{Y}^t = \mathscr{Y}^t \mathscr{C}^t = \mathscr{X}^t$ (and thus $\mathscr{X}^t \mathscr{Y}^t = \mathscr{Y}^t \mathscr{X}^t$), and

$$\Sigma_{\mathscr{A}}^t = \lambda I + \sum_{\tau=0}^{t-1} \mathscr{X}^\tau, \qquad \Sigma_{\mathscr{B}}^t = \lambda I + \sum_{\tau=0}^{t-1} \mathscr{Y}^\tau.$$

From Assumption 5, we obtain, for all $t \in [T]$, $(\mathscr{C}^t)^{-1}$ exists and

$$\|(\mathscr{C}^t)^{-1}\| \le \left\| \left( \sum_{n=0}^{N^t-1} \sum_{h=0}^{H_n^t-1} \phi_{x_{h,n}^t} \phi_{x_{h,n}^t}^\dagger \right)^{-1} \right\| \le \left\| \left( \sum_{n=0}^{N^t-1} \phi_{x_{0,n}^t} \phi_{x_{0,n}^t}^\dagger \right)^{-1} \right\| \le C^{-1}.$$

Therefore, using Lemma F.1 by substituting $I$ and $\mathscr{C}$ to $A$, $\lambda I$ and $\mathscr{Y}$ to $B$, it follows that

$$\left\| (\Sigma_{\mathscr{A}}^t)^{-\frac{1}{2}} \Sigma_{\mathscr{B}}^t (\Sigma_{\mathscr{A}}^t)^{-\frac{1}{2}} \right\| \le \max\left\{ 1, \max_{\tau \in [t]} \{\|(\mathscr{C}^\tau)^{-1}\|\} \right\} \le 1 + C^{-1},$$

and

$$\left\| (\Sigma_{\mathscr{B}}^t)^{\frac{1}{2}} \left( M - \overline{M}^t \right) \right\|^2 = \left\| (\Sigma_{\mathscr{B}}^t)^{\frac{1}{2}} (\Sigma_{\mathscr{A}}^t)^{-\frac{1}{2}} (\Sigma_{\mathscr{A}}^t)^{\frac{1}{2}} \left( M - \overline{M}^t \right) \right\|^2$$
$$\le \left\| (\Sigma_{\mathscr{A}}^t)^{\frac{1}{2}} \left( M - \overline{M}^t \right) \right\|^2 \left\| (\Sigma_{\mathscr{A}}^t)^{-\frac{1}{2}} (\Sigma_{\mathscr{B}}^t)^{\frac{1}{2}} \right\|^2 = \left\| (\Sigma_{\mathscr{A}}^t)^{-\frac{1}{2}} \Sigma_{\mathscr{B}}^t (\Sigma_{\mathscr{A}}^t)^{-\frac{1}{2}} \right\| \left\| (\Sigma_{\mathscr{A}}^t)^{\frac{1}{2}} \left( M - \overline{M}^t \right) \right\|^2$$
$$\le (1 + C^{-1}) \left\| (\Sigma_{\mathscr{A}}^t)^{\frac{1}{2}} \left( M - \overline{M}^t \right) \right\|^2.$$

Here, the second equality used $\|\mathscr{A}\|^2 = \|\mathscr{A}\mathscr{A}^\dagger\|$. $\qquad\square$

Now, let $\mathcal{E}^t$ be the event $M^\star \in \mathrm{BALL}_M^t$. Assume $\bigcap_{t=0}^{T-1} \mathcal{E}^t$. Then, using Assumption 4,

$$\left( \Lambda[\mathscr{K}(\Theta^t)] + J^{\Theta^t}(X_0^t; c^t) \right) - \left( \Lambda[\mathscr{K}(\Theta^{\star t})] + J^{\Theta^{\star t}}(X_0^t; c^t) \right)$$
$$= \left( \Lambda[\mathscr{K}(\Theta^t)] - \Lambda[\mathscr{K}(\Theta^{\star t})] \right) + \left( J^{\Theta^t}(X_0^t; c^t) - J^{\Theta^{\star t}}(X_0^t; c^t) \right)$$
$$\le \left( \Lambda[\mathscr{K}(\Theta^t)] - \Lambda[\mathscr{B}^t \circ \hat{M}^t] \right) + \left( J^{\Theta^t}(X_0^t; c^t) - J^{\Theta^t}\left( X_0^t; \hat{M}^t; c^t \right) \right)$$
$$\le \left( \left| \Lambda[\mathscr{K}(\Theta^t)] - \Lambda[\mathscr{B}^t \circ \hat{M}^t] \right| \right) + \left( J^{\Theta^t}(X_0^t; c^t) - J^{\Theta^t}\left( X_0^t; \hat{M}^t; c^t \right) \right)$$
$$\le \underbrace{\min\left\{ L \cdot \max\left\{ \left\| \mathscr{B}^t \left( M^\star - \hat{M}^t \right) \right\|^2, \left\| \mathscr{B}^t \left( M^\star - \hat{M}^t \right) \right\|^\alpha \right\}, 2\Lambda_{\max} \right\}}_{\text{term}_1} + \underbrace{\left( J^{\Theta^t}(X_0^t; c^t) - J^{\Theta^t}\left( X_0^t; \hat{M}^t; c^t \right) \right)}_{\text{term}_2}.$$

$$(\text{F.1})$$

Here, the first inequality follows because we assumed $\mathcal{E}^t$ and because the algorithm selects $\hat{M}^t$ and $\Theta^t$ such that

$$\Lambda[\mathscr{B}^t \circ \hat{M}^t] + J^{\Theta^t}\left( X_0^t; \hat{M}^t; c^t \right) \le \Lambda[\mathscr{B}^t \circ M] + J^\Theta\left( X_0^t; M; c^t \right)$$

for any $M \in \mathrm{BALL}_M^t$ and for any $\Theta \in \Pi$. The third inequality follows from Assumption 4.

Using Lemma F.2, we have

$$\left\| \mathscr{B}^t \left( M^\star - \hat{M}^t \right) \right\| \le \left\| (\Sigma_{\mathscr{B}}^t)^{\frac{1}{2}} \left( M^\star - \hat{M}^t \right) \right\| \left\| \mathscr{B}^t (\Sigma_{\mathscr{B}}^t)^{-\frac{1}{2}} \right\|$$
$$\le \sqrt{(1 + C^{-1})} \left\| (\Sigma_{\mathscr{A}}^t)^{\frac{1}{2}} \left( M^\star - \hat{M}^t \right) \right\| \left\| \mathscr{B}^t (\Sigma_{\mathscr{B}}^t)^{-\frac{1}{2}} \right\|$$
$$\le \sqrt{(1 + C^{-1})} \left( \left\| (\Sigma_{\mathscr{A}}^t)^{\frac{1}{2}} \left( M^\star - \overline{M}^t \right) \right\| + \left\| (\Sigma_{\mathscr{A}}^t)^{\frac{1}{2}} \left( \overline{M}^t - \hat{M}^t \right) \right\| \right) \left\| \mathscr{B}^t (\Sigma_{\mathscr{B}}^t)^{-\frac{1}{2}} \right\|$$
$$\le 2\sqrt{(1 + C^{-1}) \beta_M^t} \left\| \mathscr{B}^t (\Sigma_{\mathscr{B}}^t)^{-\frac{1}{2}} \right\| \qquad (\because \mathcal{E}^t).$$

Therefore, if $\mathcal{E}^t$, it follows that

$$\text{term}_1 \leq \min\left\{ L\left\{4(1+C^{-1})\beta_M^t + 1\right\} \max\left\{\left\|\mathscr{B}^t(\Sigma_\mathscr{B}^t)^{-\frac{1}{2}}\right\|^2, \left\|\mathscr{B}^t(\Sigma_\mathscr{B}^t)^{-\frac{1}{2}}\right\|^\alpha\right\}, 2\Lambda_{\max}\right\}. \tag{F.2}$$

Then, we use the following lemma which is an extension of (Kakade et al., 2020, Lemman B.6) to our Hölder condition.

**Lemma F.3.** *For any sequence of $\mathscr{B}^t$ and for any $\alpha \in (0,1]$, we have*

$$\sum_{t=0}^{T-1} \min\left\{\left\|\mathscr{B}^t(\Sigma_\mathscr{B}^t)^{-\frac{1}{2}}\right\|^{2\alpha}, 1\right\} \leq 2T^{1-\alpha}\left[1 + \log\left(\frac{\det\left(\Sigma_\mathscr{B}^T\right)}{\det\left(\Sigma_\mathscr{B}^0\right)}\right)\right].$$

*Proof.* Using $x \leq 2\log(1+x)$ for $x \in [0,1]$,

$$\sum_{t=0}^{T-1} \min\left\{\left\|\mathscr{B}^t(\Sigma_\mathscr{B}^t)^{-\frac{1}{2}}\right\|^{2\alpha}, 1\right\} \leq \sum_{t=0}^{T-1} \min\left\{\left\|\mathscr{B}^t(\Sigma_\mathscr{B}^t)^{-\frac{1}{2}}\right\|_{\text{HS}}^{2\alpha}, 1\right\} \quad (\because \|\mathscr{A}\| \leq \|\mathscr{A}\|_{\text{HS}})$$

$$= \sum_{t=0}^{T-1}\left(\min\left\{\left\|\mathscr{B}^t(\Sigma_\mathscr{B}^t)^{-\frac{1}{2}}\right\|_{\text{HS}}^2, 1\right\}\right)^\alpha \leq \sum_{t=0}^{T-1}\left[2\log\left(1 + \left\|\mathscr{B}^t(\Sigma_\mathscr{B}^t)^{-\frac{1}{2}}\right\|_{\text{HS}}^2\right)\right]^\alpha$$

$$= \sum_{t=0}^{T-1}\left[2\log\left(1 + \text{tr}\left\{(\Sigma_\mathscr{B}^t)^{-\frac{1}{2}}\mathscr{B}^{t\dagger}\mathscr{B}^t(\Sigma_\mathscr{B}^t)^{-\frac{1}{2}}\right\}\right)\right]^\alpha$$

$$\leq 2^\alpha \sum_{t=0}^{T-1}\left[\log\det\left(I + (\Sigma_\mathscr{B}^t)^{-\frac{1}{2}}\mathscr{B}^{t\dagger}\mathscr{B}^t(\Sigma_\mathscr{B}^t)^{-\frac{1}{2}}\right)\right]^\alpha$$

$$\leq 2^\alpha T^{1-\alpha}\left[\sum_{t=0}^{T-1}\log\det\left(I + (\Sigma_\mathscr{B}^t)^{-\frac{1}{2}}\mathscr{B}^{t\dagger}\mathscr{B}^t(\Sigma_\mathscr{B}^t)^{-\frac{1}{2}}\right)\right]^\alpha$$

$$\leq 2^\alpha T^{1-\alpha}\left[\sum_{t=0}^{T-1}\left(\log\det\left(\Sigma_\mathscr{B}^{t+1}\right) - \log\det\left(\Sigma_\mathscr{B}^t\right)\right)\right]^\alpha \leq 2T^{1-\alpha}\left[\log\left(\frac{\det\left(\Sigma_\mathscr{B}^T\right)}{\det\left(\Sigma_\mathscr{B}^0\right)}\right)\right]^\alpha$$

$$\leq 2T^{1-\alpha}\left(1 + \log\left(\frac{\det\left(\Sigma_\mathscr{B}^T\right)}{\det\left(\Sigma_\mathscr{B}^0\right)}\right)\right).$$

$\square$

Here, the fifth inequality follows from (Grafakos, 2008, Exercise 1.1.4).

**Corollary F.4.** *For any sequence of $\mathscr{B}^t$ and for any $\alpha \in (0,1]$, we have*

$$\sum_{t=0}^{T-1} \min\left\{\max\left\{\left\|\mathscr{B}^t(\Sigma_\mathscr{B}^t)^{-\frac{1}{2}}\right\|^4, \left\|\mathscr{B}^t(\Sigma_\mathscr{B}^t)^{-\frac{1}{2}}\right\|^{2\alpha}\right\}, 1\right\} \leq 2T^{1-\alpha}\left[1 + \log\left(\frac{\det\left(\Sigma_\mathscr{B}^T\right)}{\det\left(\Sigma_\mathscr{B}^0\right)}\right)\right].$$

From (F.2) and Corollary F.4, we obtain

$$
\mathbb{E}\left[\sum_{t=0}^{T-1}\mathrm{term}_1\,\middle|\,\bigcap_{t=0}^{T-1}\mathcal{E}^t\right]
$$

$$
\leq \mathbb{E}\left[\sum_{t=0}^{T-1}\min\left\{L\left\{4(1+C^{-1})\beta_M^t+1\right\}\max\left\{\left\|\mathscr{B}^t(\Sigma_\mathscr{B}^t)^{-\frac{1}{2}}\right\|^2,\left\|\mathscr{B}^t(\Sigma_\mathscr{B}^t)^{-\frac{1}{2}}\right\|^\alpha\right\},2\Lambda_{\max}\right\}\right]
$$

$$
\leq \mathbb{E}\left[\sum_{t=0}^{T-1}\left\{L\left\{4(1+C^{-1})\beta_M^t+1\right\}+2\Lambda_{\max}\right\}\min\left\{\max\left\{\left\|\mathscr{B}^t(\Sigma_\mathscr{B}^t)^{-\frac{1}{2}}\right\|^2,\left\|\mathscr{B}^t(\Sigma_\mathscr{B}^t)^{-\frac{1}{2}}\right\|^\alpha\right\},1\right\}\right]
$$

$$
\leq \sum_{t=0}^{T-1}\sqrt{\mathbb{E}\left[\,[L\{4(1+C^{-1})\beta_M^t+1\}+2\Lambda_{\max}]^2\right]}\sqrt{\mathbb{E}\left[\min\left\{\max\left\{\left\|\mathscr{B}^t(\Sigma_\mathscr{B}^t)^{-\frac{1}{2}}\right\|^4,\left\|\mathscr{B}^t(\Sigma_\mathscr{B}^t)^{-\frac{1}{2}}\right\|^{2\alpha}\right\},1\right\}\right]}
$$

$$
\leq \sqrt{\sum_{t=0}^{T}\mathbb{E}\left[\,[L\{4(1+C^{-1})\beta_M^t+1\}+2\Lambda_{\max}]^2\right]}\sqrt{\mathbb{E}\left[\sum_{t=0}^{T-1}\min\left\{\max\left\{\left\|\mathscr{B}^t(\Sigma_\mathscr{B}^t)^{-\frac{1}{2}}\right\|^4,\left\|\mathscr{B}^t(\Sigma_\mathscr{B}^t)^{-\frac{1}{2}}\right\|^{2\alpha}\right\},1\right\}\right]}
$$

$$
\leq \sqrt{T\cdot\mathtt{value}}\sqrt{T^{1-\alpha}(2+\gamma_{T,\mathscr{B}}(\lambda))}. \tag{F.3}
$$

Here, the first inequality is due to (F.2); the third inequality uses $\mathbb{E}[ab]\leq\sqrt{\mathbb{E}[a^2]\mathbb{E}[b^2]}$; the forth inequality uses the Cauchy-Schwartz inequality; the last is from Corollary F.4. Also,

$$
\mathtt{value} := 16L^2(1+C^{-1})^2\mathbb{E}[(\beta_M^T)^2] + (8L^2+16\Lambda_{\max}L)(1+C^{-1})\mathbb{E}[\beta_M^T] + 4\Lambda_{\max}L + L^2 + 4\Lambda_{\max}^2
$$

$$
\leq C'\left\{L^2(1+C^{-1})^2\mathbb{E}[(\beta_M^T)^2] + (L^2+\Lambda_{\max}L)(1+C^{-1})\mathbb{E}[\beta_M^T] + \Lambda_{\max}^2 + L^2\right\},
$$

for some constant $C'$.

Next, we turn to bound the latter term $\mathrm{term}_2$ of (F.1); our analysis is based on that of (Kakade et al., 2020). Simple calculations show that

$$
M^\star - \overline{M}^t = \lambda(\Sigma_\mathscr{A}^t)^{-1}M^\star - (\Sigma_\mathscr{A}^t)^{-1}\sum_{\tau=0}^{t-1}\sum_{n=0}^{N^\tau-1}\sum_{h=0}^{H_n^\tau-1}\mathscr{A}_{h,n}^{\tau}{}^{\dagger}\epsilon_{h,n}^\tau,
$$

where $\epsilon_{h,n}^\tau$ is the sampled noise at $\tau$-th episode, $h$-th timestep, and $n$-th trajectory. Now, by introducing a Hilbert space containing an operator of $\mathcal{L}(\mathcal{L}(\mathcal{H};\mathcal{H}');\mathbb{R})$, which is a Hilbert-Schmidt operator, because of Assumption 1, we can apply Lemma C.4 in (Kakade et al., 2020) to our problem too. Therefore, with probability at least $1-\delta_t$, it holds that

$$
\left\|(\Sigma_\mathscr{A}^t)^{\frac{1}{2}}\left(M-\overline{M}^t\right)\right\|^2 \leq \lambda\|M^\star\|^2 + \sigma^2(8d_\phi\log(5) + 8\log(\det(\Sigma_\mathscr{A}^t)\det(\Sigma_\mathscr{A}^0)^{-1})/\delta_t),
$$

and properly choosing $\delta_t$ leads to $\beta_M^t$ defined in Section B, and we obtain the result

$$
\Pr\left(\bigcup_{t=0}^{T-1}\overline{\mathcal{E}^t}\right) \leq \frac{1}{2}.
$$

In our algorithm, transition data are chosen from any initial states and the horizon lengths vary; however, slight modification of the analysis of $\mathrm{LC}^3$ will give the following lemma.

**Lemma F.5** (Modified version of Theorem 3.2 in (Kakade et al., 2020)). *Suppose Assumptions 1, 2, 3, 5, and 6 hold. Then, the term* $\mathrm{term}_2$ *is bounded by*

$$
\mathbb{E}\left[\sum_{t=0}^{T-1}\mathrm{term}_2\,\middle|\,\bigcap_{t=0}^{T-1}\mathcal{E}^t\right]
$$

$$
\leq \sqrt{HV_{\max}}\sqrt{64T(d_\phi + \log(T) + \gamma_{T,\mathscr{A}}(\lambda) + H)}\sqrt{\gamma_{T,\mathscr{A}}(\lambda)}.
$$

Combining all of the above results, we prove Theorem 4.10:

*Proof of Theorem 4.10.* Using (F.3) (which requires Assumptions 1, 2, 4, and 5), Lemma F.5 (which requires Assumptions 1, 2, 3, 5, and 6), and $\Pr\left(\bigcup_{t=0}^{T-1} \overline{\mathcal{E}^t}\right) \leq \frac{1}{2}$, it follows that

$$
\begin{aligned}
&\mathbb{E}_{\mathrm{KS-LC^3}}\left[\mathrm{REGRET}_T\right] \\
&\leq \sqrt{T\left\{C'\left\{L^2(1+C^{-1})^2\mathbb{E}[(\beta_M^T)^2] + (L^2 + \Lambda_{\max}L)(1+C^{-1})\mathbb{E}[\beta_M^T] + \Lambda_{\max}^2 + L^2\right\}\right\}}\sqrt{T^{1-\alpha}(2+\gamma_{T,\mathscr{B}}(\lambda))} \\
&\quad + \sqrt{HV_{\max}}\sqrt{64T(d_\phi + \log(T) + \gamma_{T,\mathscr{A}}(\lambda) + H)}\sqrt{\gamma_{T,\mathscr{A}}(\lambda)} \\
&\quad + \frac{1}{2}\cdot(\Lambda_{\max} + \sqrt{V_{\max}}) \\
&\leq C_1 T^{1-\frac{\alpha}{2}}(\tilde{d}_{T,1} + \tilde{d}_{T,2}),
\end{aligned}
$$

for some absolute constant $C_1$. Therefore, the theorem is proved. $\qquad\square$

## G   Reduction to eigenstructure assignment problem for linear systems

To see how our system model studied in (4.2) reduces to linear system case, take $\mathbb{R}^{d_\mathcal{X}}$ as $\mathcal{H}_0$ with the canonical basis (i.e., $\boldsymbol{\phi}_x = x$) and let

$$
\Phi(\Theta) = [I_{d_\mathcal{X}} \otimes k_1^\top, I_{d_\mathcal{X}} \otimes k_2^\top, \ldots, I_{d_\mathcal{X}} \otimes k_{d_\mathcal{X}}^\top, I_{d_\mathcal{X}}]^\top,
$$

where the feedback matrix is given by $K := [k_1, k_2, \ldots, k_{d_\mathcal{X}}]$, and let

$$
M^* = \left[[\boldsymbol{b}_1, a_1]^\top, \ldots, [\boldsymbol{b}_{d_\mathcal{X}}, a_{d_\mathcal{X}}]^\top\right],
$$

where the entries of the row vector $\boldsymbol{b}_i$ are all zero except for the entries from the index $(i-1)d_\mathcal{X}d_\mathcal{U}+1$ to the index $id_\mathcal{X}d_\mathcal{U}$ given by $\mathrm{vec}\left(B^\top\right)$, and $A = [a_1^\top, a_2^\top, \ldots, a_{d_\mathcal{X}}^\top]$.

# H   Setups and results of simulations in the main body

Throughout the main body of this paper, we used the following version of Julia; for each experiment, the running time was less than around 10 minutes.

```
Julia Version 1.5.3
Platform Info:
OS: Linux (x86_64-pc-linux-gnu)
CPU: AMD Ryzen Threadripper 3990X 64-Core Processor
WORD_SIZE: 64
LIBM: libopenlibm
LLVM: libLLVM-9.0.1 (ORCJIT, znver2)
Environment:
JULIA_NUM_THREADS = 12
```

The licenses of Julia, OpenAI Gym, DeepMind Control Suite, Lyceum, and MuJoCo, are [The MIT License; Copyright (c) 2009-2021: Jeff Bezanson, Stefan Karpinski, Viral B. Shah, and other contributors: https://github.com/JuliaLang/julia/contributors], [The MIT License; Copyright (c) 2016 OpenAI (https://openai.com)], [Apache License Version 2.0, January 2004 http://www.apache.org/licenses/], [The MIT License; Copyright (c) 2019 Colin Summers, The Contributors of Lyceum], and [MuJoCo Pro Lab license], respectively.

In this section, we provide simulation setups, including the details of environments (see also Figure 9) and parameter settings.

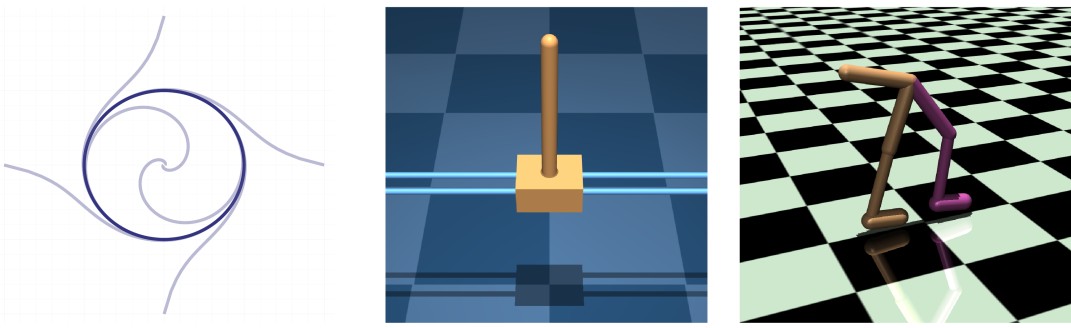

Figure 9: Illustration of some dynamical systems we have used in this work. Left: Simple limit cycle represented effectively by the Koopman modes. Middle: DeepMind Control Suite (Tassa et al., 2018) Cartpole showing stable cycle with spectral radius regularization. Right: OpenAI Gym (Brockman et al., 2016) walker2d showing simpler movement cycle when the Koopman eigenvalues are regularized.

## H.1   Cross-entropy method

Throughout, we used CEM for dynamics parameter (policy) selection to approximately solve KSNR. Here, we present the setting of CEM.

First, we prepare some fixed feature (e.g. RFFs) for $\phi$. Then, at each iteration of CEM, we generate many parameters to compute the loss (i.e., the sum of the Koopman spectrum cost and *negative* cumulative reward) by fitting the transition data generated by each parameter to the feature to estimate its Koopman operator $\mathscr{A}$. In particular, we used the following regularized fitting:

$$\mathscr{A} = YX^\top(XX^\top + I)^{-1},$$

Table 1: Hyperparameters used for limit cycle generation.

| CEM hyperparameter | Value | Training target Koopman operator | Value |
|---|---|---|---|
| samples | 200 | training iteration | 500 |
| elite size | 20 | RFF bandwidth for $\phi$ | 3.0 |
| iteration | 50 | RFF dimension $d_\phi$ | 80 |
| planning horizon | 80 | horizon for each iteration | 80 |
| policy RFF dimension | 50 | | |
| policy RFF bandwidth | 2.0 | | |

where $Y := [\phi_{x_{h_1+1,1}}, \phi_{x_{h_2+1,2}}, \ldots, \phi_{x_{h_n+1,n}}]$ and $X := [\phi_{x_{h_1,1}}, \phi_{x_{h_2,2}}, \ldots, \phi_{x_{h_n,n}}]$.

If the feature spans a Koopman invariant space and the deterministic dynamical systems are considered, and if no regularization (i.e., the identity matrix $I$) is used, any sufficiently rich trajectory data may be used to exactly compute $\mathscr{K}(\Theta)$ for $\Theta$. However, in practice, the estimate depends on transition data although the regularization mitigates this effect. In our simulations, at each iteration, we randomly reset the initial state according to some distribution, and computed loss for each parameter generating trajectory starting from that state.

## H.2 Setups: imitating target behaviors through Koopman operators

The discrete-time dynamics

$$r_{h+1} = r_h + v_{r,h}\Delta t, \ \theta_{h+1} = \theta_h + v_{\theta,h}\Delta t$$

is considered and the policy returns $v_{r,h}$ and $v_{\theta,h}$ given $r_h$ and $\theta_h$. In our simulation, we used $\Delta t = 0.05$. Note the ground-truth dynamics

$$\dot{r} = r(1 - r^2), \ \dot{\theta} = 1,$$

is discretized to

$$r_{h+1} = r_h + r_h(1 - r_h^2)\Delta t, \ \theta_{h+1} = \theta_h + \Delta t.$$

Figure 10 plots the ground-truth trajectories of observations and $x$-$y$ positions.

We trained the target Koopman operator using the ground-truth dynamics with random initializations; the hyperparameters used for training are summarized in Table 1.

Then, we used CEM to select policy so that the spectrum cost is minimized; the hyperparameters are also summarized in Table 1.

We tested two forms of the spectrum cost; $\Lambda_1(\mathscr{A}) = \|\mathbf{m} - \mathbf{m}^\star\|_1$ and $\Lambda_2(\mathscr{A}) = \|\mathscr{A} - \mathscr{A}^\star\|_{\mathrm{HS}}^2$. The resulting trajectories are plotted in Figure 11 and 12, respectively. It is interesting to observe that the top mode imitation successfully converged to the desirable limit cycle while Frobenius norm imitation did not. Intuitively, the top mode imitation focuses more on reconstructing the practically and physically meaningful behavior while minimizing the error on the Frobenius norm has no immediately clear physical meaning.

## H.3 Setups: Generating stable loops (Cartpole)

We used DeepMind Control Suite Cartpole environment with modifications; specifically, we extended the cart rail to $[-100, 100]$ from the original length $[-5, 5]$ to deal with divergent behaviors. Also, we used a combination of linear and RFF features; the first elements of the feature are simply the observation (state) vector, and the rest are Gaussian RFFs. That way, we found divergent behaviors were well-captured in terms of spectral radius. The hyperparemeters used for CEM are summarized in Table 2.

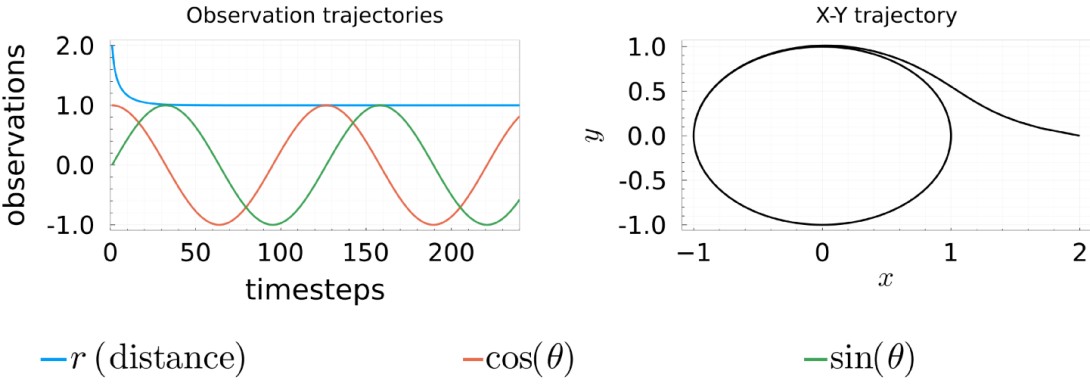

Figure 10: The ground-truth trajectory of the limit cycle $\dot{r} = r(1 - r^2)$, $\dot{\theta} = 1$. Left: Observations $r$, $\cos(\theta)$, and $\sin(\theta)$. Right: $x$-$y$ positions.

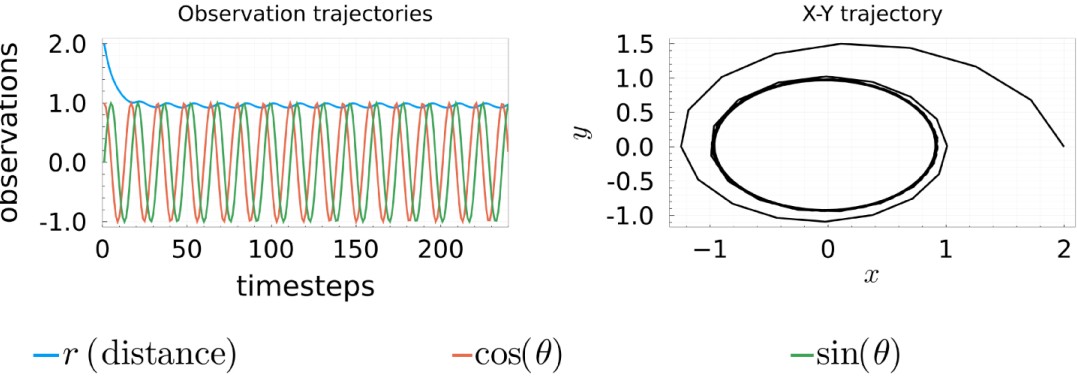

Figure 11: The trajectory generated by RFF policies that minimize $\Lambda(\mathscr{A}) = \|\mathbf{m} - \mathbf{m}^\star\|_1$. Left: Observations $r$, $\cos(\theta)$, and $\sin(\theta)$. Right: $x$-$y$ positions.

### H.4 Setups: Generating smooth movements (Walker)

Because of the complexity of the dynamics, we used four random seeds in this simulation, namely, 100, 200, 300, and 400. We used a combination of linear and RFF features for both $\phi$ and the policy. Note, according to the work (Rajeswaran et al., 2017), linear policy is actually sufficient for some tasks for particular environments. The hyperparemeters used for CEM are summarized in Table 3.

The resulting trajectories of Walker are illustrated in Figure 13. The results are rather surprising; because we did not specify the height in reward, the dynamics with only cumulative cost showed rolling behavior (Up figure) to go right faster most of the time. On the other hand, when the spectrum cost was used, the hopping behavior (Down figure) emerged. Indeed this hopping behavior moves only one or two joints most of the time while fixing other joints, which leads to lower (absolute values of) eigenvalues.

The eigenspectrums of the resulting dynamics with/without the spectrum cost are plotted in Figure 14. In fact, it is observed that the dynamics when the spectrum cost was used showed consistently lower (absolute values of) eigenvalues; for the hopping behavior, most of the joint angles converged to some values and stayed there.

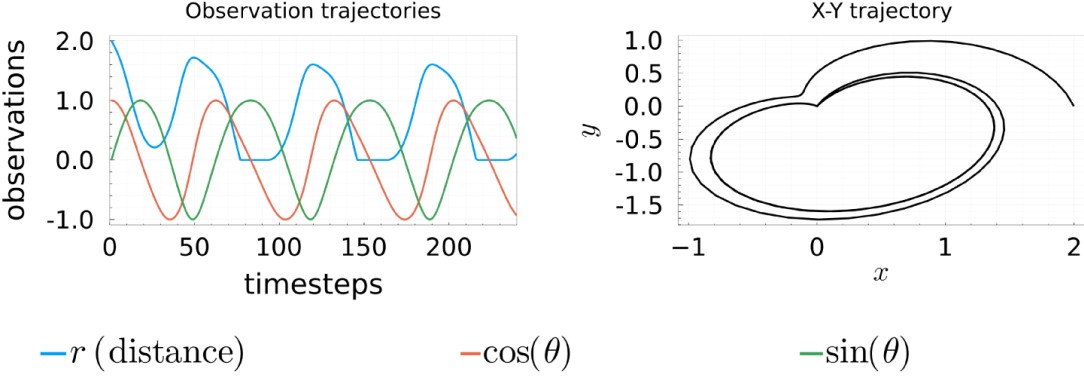

Figure 12: The trajectory generated by RFF policies that minimize $\Lambda(\mathscr{A}) = \|\mathscr{A} - \mathscr{A}^\star\|_{\mathrm{HS}}^2$. Left: Observations $r$, $\cos(\theta)$, and $\sin(\theta)$. Right: $x$-$y$ positions.

Table 2: Hyperparameters used for stable loop generation.

| Hyperparameters | Value | Hyperparameters | Value |
|---|---|---|---|
| samples | 200 | elite size | 20 |
| iteration | 100 | planning horizon | 100 |
| dimension $d_\phi$ | 50 | RFF bandwidth for $\phi$ | 2.0 |
| policy RFF dimension | 100 | policy RFF bandwidth | 2.0 |

---

**Algorithm 2** Practical Algorithm for KS-LC$^3$

---

**Require:** Parameter set $\Pi$; prior parameter $\lambda$; covariance scale $\iota \in \mathbb{R}_{\geq 0}$.

1: Prior distribution over $M'$ is given by $(\Sigma_{M'}^0)^{-1}$ where $\Sigma_{M'}^0 := \lambda I$

2: **for** $t = 0 \ldots T - 1$ **do**

3:     Adversary chooses $X_0^t$.

4:     Sample $\hat{M}'^t$ from $\mathcal{N}\left(\overline{M'}^t, (\Sigma_{M'}^t)^{-1}\right)$

5:     Solve $\Theta^t = \arg\min_{\Theta \in \Pi} \Lambda[\hat{\mathscr{K}}^t(\Theta)] + J^\Theta\left(X_0^t; \hat{M}'^t; c^t\right)$ (e.g., using CEM)

6:     Under the dynamics $\mathcal{F}^{\Theta^t}$, sample transition data $\tau^t := \{\tau_n^t\}_{n=0}^{N^t-1}$, where $\tau_n^t := \{x_{h,n}^t\}_{h=0}^{H_n^t}$

7:     Update $\Sigma_{M'}^t$

8: **end for**

---

## H.5 Setups: Koopman-Spectrum LC$^3$

**Decomposable kernels case** We explain how to reduce the memory size in a special decomposable kernel case. Assume that we employ decomposable kernel with $B = I$ for simplicity. Also, suppose $\Psi(\Theta) \in \mathbb{R}^{d_\Psi \times d_\phi}$ is of finite dimension. For such a case, the dimension $d_\Psi = d_\zeta \cdot d_\phi$ where $d_\zeta$ is the dimension of $\mathcal{H}''$; however, we do not need to store a covariance matrix of size $d_\phi^2 d_\zeta \times d_\phi^2 d_\zeta$ but only require to update a matrix of size $d_\phi d_\zeta \times d_\phi d_\zeta$ which significantly reduces the memory size. Specifically, we consider the model $\phi_{x_{h+1}} = M'(\phi_{x_h} \otimes \zeta(\Theta))$; then using $M'' := \mathrm{reshape}(M', d_\phi, d_\zeta, d_\phi)$, we obtain $\mathscr{K}(\Theta) = [M''[:,:,1]\zeta(\Theta), \ldots, M''[:,:,d_\phi]\zeta(\Theta)]$. Here, $\zeta$ is the realization of $\zeta(\Theta)$ over some basis. Now, we note that the dimension of $\phi_{x_h} \otimes \zeta(\Theta)$ is $d_\zeta \cdot d_\phi$. Our practical (Thompson sampling version) algorithm is thus given by Algorithm 2. In the algorithm, we used $\hat{\mathscr{K}}^t(\Theta) = [\hat{M}''^t[:,:,1]\zeta(\Theta), \ldots, \hat{M}''^t[:,:,d_\phi]\zeta(\Theta)]$, where $\hat{M}''^t := \mathrm{reshape}(\hat{M}'^t, d_\phi, d_\zeta, d_\phi)$.

Table 3: Hyperparameters used for Walker.

| Hyperparameters | Value | Hyperparameters | Value |
|---|---|---|---|
| samples | 300 | elite size | 20 |
| iteration | 50 | planning horizon | 300 |
| dimension of $d_\phi$ | 200 | RFF bandwidth for $\phi$ | 5.0 |
| policy RFF dimension | 300 | policy RFF bandwidth | 30.0 |

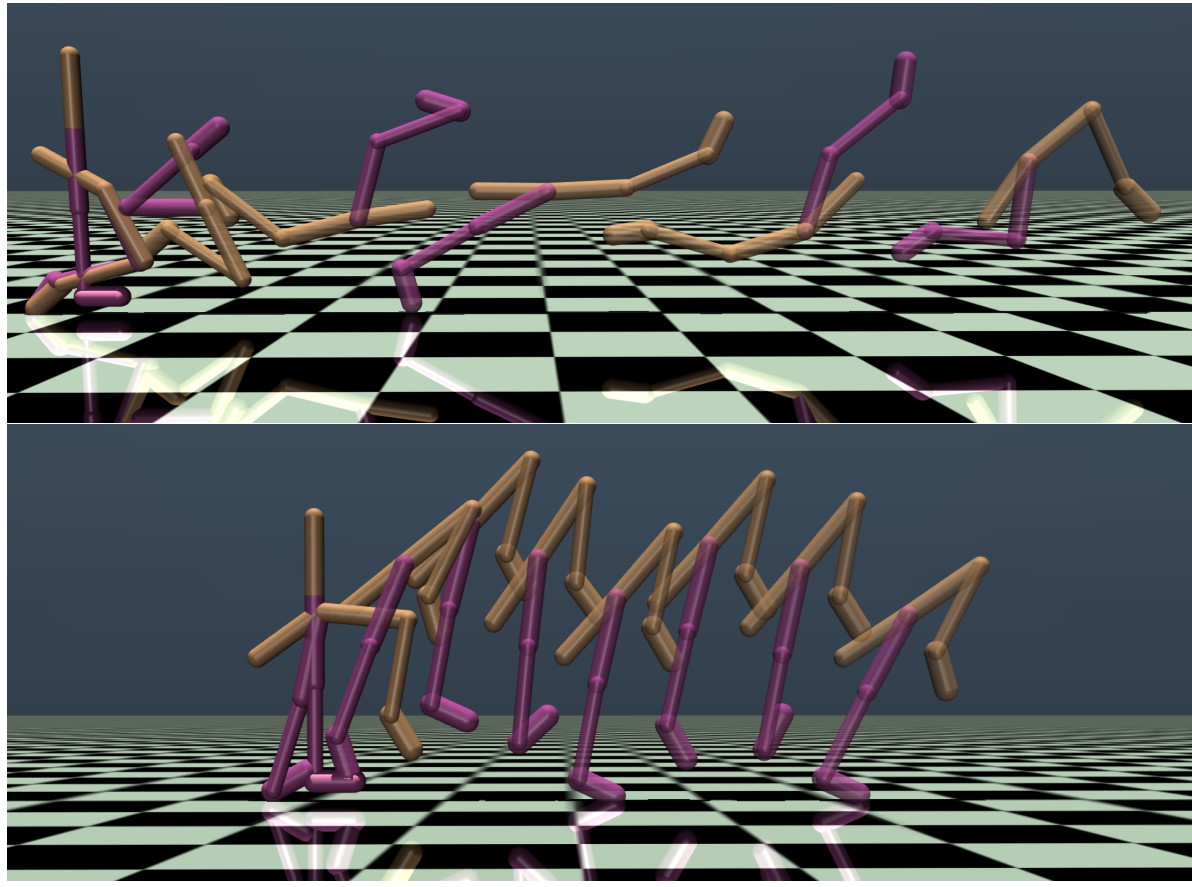

Figure 13: Walker trajectories visualized via Lyceum. Up: When only (single-step) reward $v - 0.001\|a\|^2_{\mathbb{R}^6}$ is used, showing rolling behavior. Down: When the spectrum cost $\Lambda(\mathscr{A}) = 5\sum_{i=1}^{d_\phi}|\lambda_i(\mathscr{A})|$ is used together with the reward, showing simple hopping behavior.

**Simple linear system experiment** The hyperparameters used for KS-LC$^3$ in the simple linear system experiment are summarized in Table 4.

**Cartpole pretraining policies** For training three policies, we used Model Predictive Path Integral Control (MPPI) (Williams et al., 2017) with the rewards $(p+0.3)^2$, $(p-0.3)^2$, $(v+1.5)^2$, and $(v-1.5)^2$, where $p$ is the cart position and $v$ is the cart velocity. Also, for all of the cases, the penalty $-100$ is added when $\cos(\theta) < 0$, where $\theta$ is the pole angle, which aims at preventing the pole from falling down.

Because we need to have one more state dimension to specify which reward to generate, we used the analytical model of cartpole specified in OpenAI Gym.

Starting from random initial state, we first use MPPI to move to $p = -0.3$, then from there move to $p = 0.3$; and we learn an RFF policy for this movement along with the Koopman operator. Then, we randomly

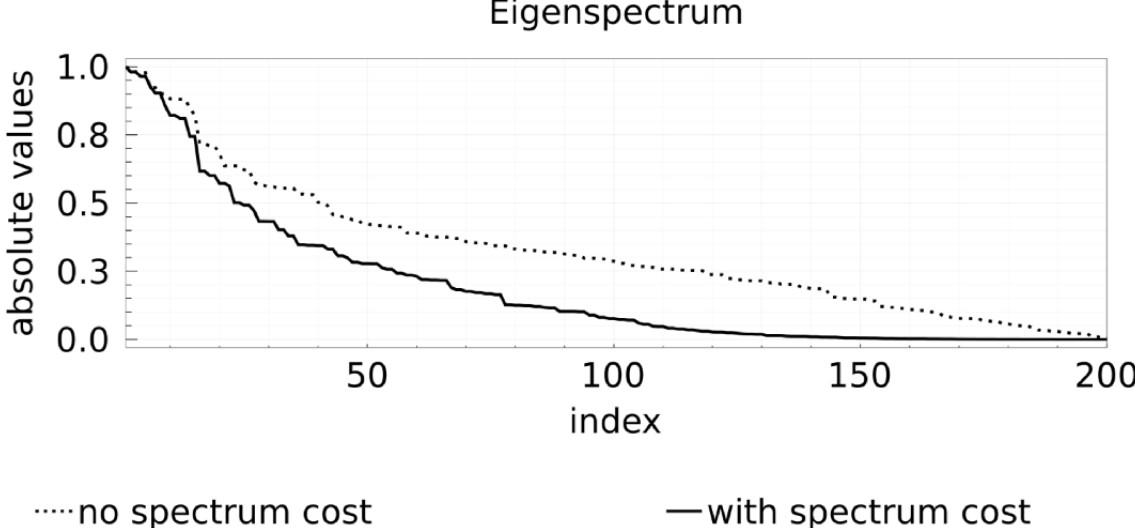

Figure 14: Eigenspectrums showing absolute values of eigenvalues for the dynamics with/without the spectrum cost.

Table 4: Hyperparameters used for KS-LC$^3$ in the simple linear case.

| Hyperparameters | Value | Hyperparameters | Value |
|---|---|---|---|
| prior parameter $\lambda$ | 0.05 | covariance scale $\iota$ | 0.0001 |
| planning horizon | 50 | | |

initialized the state to accelerate to $v = -1.5$, followed by a random initialization again to accelerate to $v = 1.5$; we then learned two policies for those two movements. We used the planning horizons of 100 for every movement except for the movement going to $p = 0.3$ where we used 120 because it is following the previous movement. We repeated this for 20 iterations.

The parameter space $\Pi$ is a space of linear combinations of those three policies. We summarized the hyperparemeters used for MPPI/pretraining in Table 5.

**Cartpole learning**  For learning, we used four random seeds, namely, 100, 200, 300, and 400. The estimated spectrum cost curve represents the cost $\Lambda[\hat{\mathscr{K}}^t(\Theta^t)]$. The hyperparameters used for CEM and KS-LC$^3$ are summarized in Table 6. We note that for KS-LC$^3$ we added additional cost on the policy parameter exceeding its $\ell_\infty$ norm above 2.0.

Table 5: Hyperparameters of MPPI and pretraining.

| MPPI hyperparameters | Value | Pretraining hyperparameters | Value |
|---|---|---|---|
| variance of controls | $0.4^2$ | iteration | 20 |
| temperature parameter | 0.1 | policy RFF dimension | 2000 |
| planning horizon | 100/120 | policy RFF bandwidth | 1.5 |
| number of planning samples | 524 | dimension of $d_\phi$ | 60 |
| | | RFF bandwidth for $\phi$ | 1.5 |

Table 6: Hyperparameters used for CEM/KS-LC$^3$.

| Hyperparameters for CEM | Value | Hyperparameters for KS-LC$^3$ | Value |
|---|---|---|---|
| samples | 200 | dimension of $d_\zeta$ | 50 |
| elite size | 20 | RFF bandwidth for $\zeta$ | 5.0 |
| planning horizon | 500 | prior parameter $\lambda$ | 1.0 |
| iteration | 50 | covariance scale $\iota$ | 0.0001 |

## I  Further experimental analysis

In this section, we provide additional experiments. Throughout this section, we used the following version of Julia as our computational resource has changed when conducting the experiments presented in this section.

```
Julia Version 1.5.3
Platform Info:
OS: Linux (x86_64-pc-linux-gnu)
CPU: Intel(R) Xeon(R) CPU E5-2620 v3 @ 2.40GHz
WORD_SIZE: 64
LIBM: libopenlibm
LLVM: libLLVM-9.0.1 (ORCJIT, haswell)
Environment:
JULIA_NUM_THREADS = 12
```

MuJoCo version 2.0 is used (license: MuJoCo Pro Individual license).

### I.1  Variations on stable Cartpole motions

In our simulation of generating stable Cartpole motion, we have seen that it shows oscillating behavior when stability is enforced through the spectrum cost in addition to the reward that encourages the cartpole to have larger velocity. To see this phenomenon more in details, we have conducted the same experiments for different time horizons, namely, 50, 100 and 200. Here, we use $\Lambda(\mathscr{A}) = 10^5 \max(1, \rho(\mathscr{A}))$ (10 times more weight than that in the simulation experiment in the main body). The resulting velocity trajectories with/without the spectrum cost are plotted in Figure 15. Also, the angle trajectories are plotted in Figure 16, where the zero lines are the threshold for adding penalty costs. Note for the case of time horizon 200, we used more intensive CEM search whose parameters are summarized in Table 7, but could not find a policy parameter that can keep the pole straight up. Studying more sophisticated heuristic search algorithm will be an important direction of future research. From Figure 15, it is observed that the longer horizon may not indicate more oscillation "cycles". In fact, our spectrum cost only regularizes the dynamics to be stable, which may include a motion where the velocity converges to some fixed value. The spectrum radius for the cases of time horizon 50, 100 and 200 without the spectrum cost is given by 1.003, 1.00006 and 1.002, while that with the spectrum cost is 0.992, 0.999 and 0.997. While all of them show stable Koopman spectrum when the spectrum cost is used, the case for the time horizon 100 shows particularly interesting behavior. Please recall that the failure of keeping the pole straight up is not necessarily regarded as unstable dynamics over the specified state space *where the angle representation is bounded* but costs the learner within the cumulative cost term.

### I.2  Variations on smooth Walker motions

To investigate smooth motion generations studied in the main body of this paper more, we conducted additional experiments.

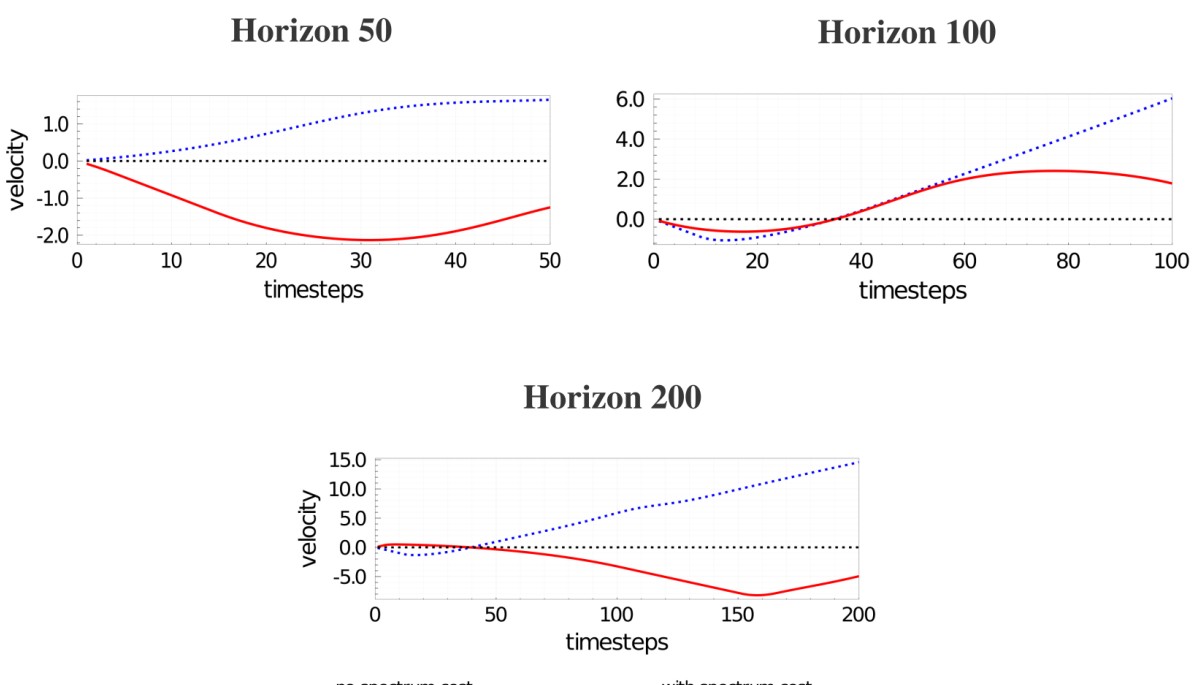

Figure 15: Velocity trajectories of Cartpole for different time horizons (50, 100 and 200) with/without the spectrum cost.

Table 7: Hyperparameters used for stabilized Cartpole for time horizon 200.

| Hyperparameters | Value | Hyperparameters | Value |
|---|---|---|---|
| samples | 200 | elite size | 20 |
| iteration | 1000 | planning horizon | 200 |
| dimension $d_\phi$ | 50 | RFF bandwidth for $\phi$ | 2.0 |
| policy RFF dimension | 300 | policy RFF bandwidth | 2.0 |

### I.2.1 Smoothness comparison with increased action cost

Especially, we also compare our KSNR for smoothness enhancements to the use of action costs in the Walker environment. In this experiment, we used the hyperparameters summarized in Table 8. We again used a combination of linear and RFF features for both $\phi$ and the policy. Recall the default immediate reward is $v - 0.001\|a\|_{\mathbb{R}^6}^2$, where $v$ is the velocity and $a$ is the action vector of dimension 6. Here, in addition to KSNR, we tested increased action cost scenarios where the immediate rewards are $v - 0.01\|a\|_{\mathbb{R}^6}^2$ and $v - 0.1\|a\|_{\mathbb{R}^6}^2$ respectively. Across the six seed runs (of seed numbers of 100, 200, 300, 400, 500, and 600), we obtained the mean of the cumulative reward and the cumulative action cost (which is computed for the trajectories using $0.001\|a\|_{\mathbb{R}^6}^2$ for all of the cases), and the mean and standard deviation of the spectrum cost, all of which are summarized in Table 9. As observed, increased action cost in our scenarios shows lower spectrum cost; while KSNR shows better cumulative reward with better spectrum cost. However, the motion generated by the increased action cost shows lower action penalty cost; which implies that the spectrum cost and the action cost have some correlation while they qualitatively prefer different motions. We also measured the smoothness by another metric than the spectrum cost itself, which is defined by

$$\text{Smoothness}(\tau) := \frac{1}{d_{\mathcal{X}} H} \sum_{h=0}^{H-1} \|x_{h+1} - x_h\|_1,$$

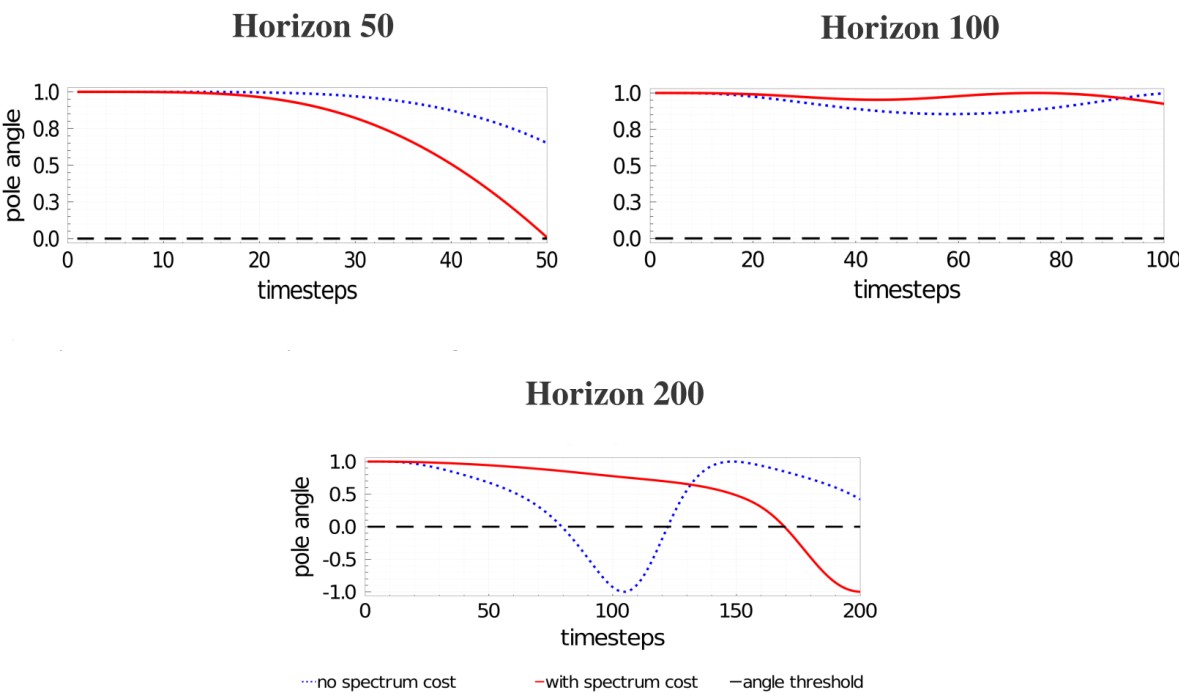

Figure 16: Angle trajectories of Cartpole for different time horizons (50, 100 and 200) with/without the spectrum cost.

Table 8: Hyperparameters used for additional Walker smoothness experiments.

| Hyperparameters | Value | Hyperparameters | Value |
|---|---|---|---|
| samples | 300 | elite size | 20 |
| iteration | 120 | planning horizon | 300 |
| dimension of $d_\phi$ | 200 | RFF bandwidth for $\phi$ | 5.0 |
| policy RFF dimension | 300 | policy RFF bandwidth | 30.0 |

where $\tau := \{x_h\}_{h=0}^H$ is a trajectory. The mean smoothness values across the runs for the motions generated by the CEM algorithms with default action cost, 10 times more action cost, 100 times more action cost, and with the spectrum cost are 0.082, 0.033, 0.007, and 0.028 respectively, and they appear to be consistent to the spectrum cost in this case. The motions are visualized in Figure 17; their joint trajectories are plotted in Figure 18 and the eigenspectrums are given in Figure 19. Note those motions are of those showing median values of the spectrum cost within the seed runs.

### I.2.2 Different feature space $\mathcal{H}_0$

Next, we examine what happens when different feature space $\mathcal{H}_0$ is employed in practice. In particular, we used a polynomial feature (spanned by $x, x^2, x^3, x^4, x^5$) for $\phi$ while the policy is again a combination of linear and RFF features. The hyperparameters are the same except for the dimension of $d_\phi$ which is now $5d_\mathcal{X}$.

Now, the spectrum costs of the generated motions of the CEM algorithms with default action cost, 10 times more action cost, 100 times more action cost are given by $232.9 \pm 13.6$, $194.5 \pm 19.5$, $152.1 \pm 42.8$, and $217.9 \pm 24.7$; and the mean of cumulative reward and action cost (penalty) of the one with spectrum cost defined over the polynomial feature space is 900.1 and 143.3. It is observed that 10 times more action cost led to lower spectrum cost in this case than KSNR while the cumulative reward of KSNR is much higher. The smoothness measure for KSNR is now 0.064, which is higher than the case with RFF features.

Table 9: Cumulative reward, cumulative action cost (penalty), and spectrum cost comparisons.

| Method (Env. setting) | Reward | Penalty | Spectrum cost (mean) | Spectrum cost (std) |
|---|---|---|---|---|
| CEM (default action cost) | 1011.5 | 177.3 | 317.0 | ±33.2 |
| CEM (×10 action cost) | 596.4 | 10.9 | 213.2 | ±50.0 |
| CEM (×100 action cost) | 63.4 | 0.5 | 88.8 | ±46.6 |
| CEM (with spectrum cost) | 737.8 | 78.1 | 186.6 | ±88.4 |

In fact, when visualizing the motion and joint trajectory Figure 20, we observe that the walker also shows similar rolling behavior but with slightly better periodicity than that one without spectrum cost. Eigenspectrums are shown in Figure 21; where we see less difference among the ones with/without spectrum cost and with ×10 action cost.

These results imply that the feature space selection influences the qualitative behavior difference in practice; please also see Remark 3.2.

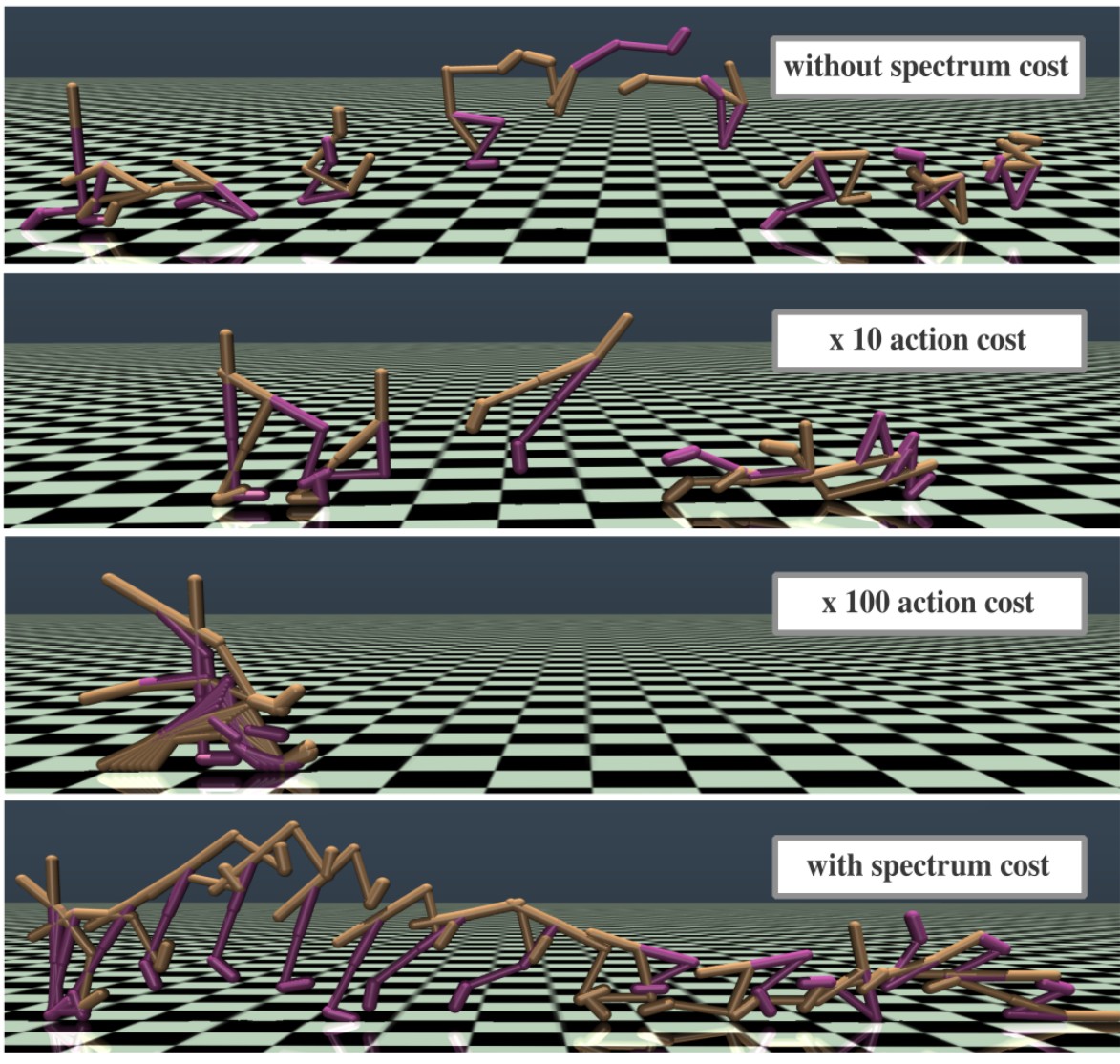

Figure 17: Visualizations of Walker motions generated by the CEM algorithm with default action cost, 10 times more action cost, 100 times more action cost, and with the spectrum cost. The motions are of those showing median values of the spectrum cost within the seed runs. It is observed that the motion generated by the one with 10 times more action cost is smooth but uses two feet to hop, which would reduce the magnitudes of actions applied to the joints. The motion generated by the one with the spectrum cost again lifts one foot and hops; this specific visualized motion then shows a bit of rotation at the last moment.

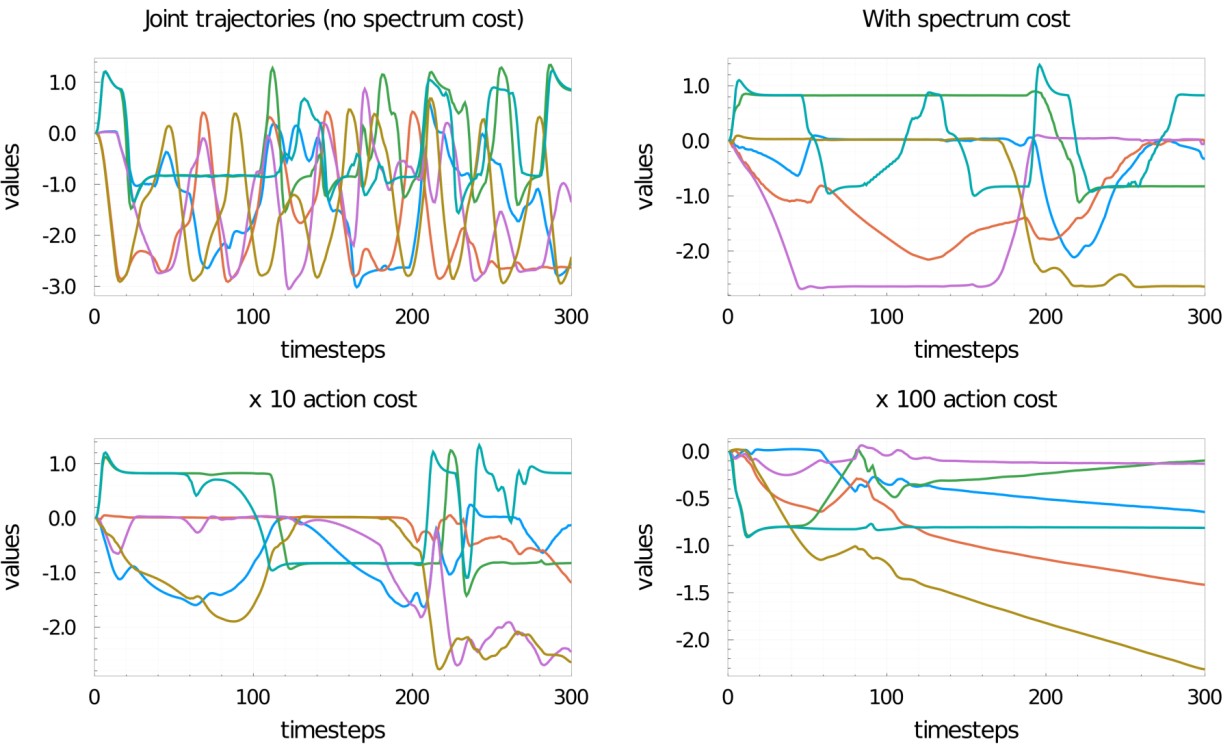

Figure 18: Joint trajectories of Walker motions generated by the CEM algorithm with default action cost, 10 times more action cost, 100 times more action cost, and with the spectrum cost. They are of those showing median values of the spectrum cost within the seed runs.

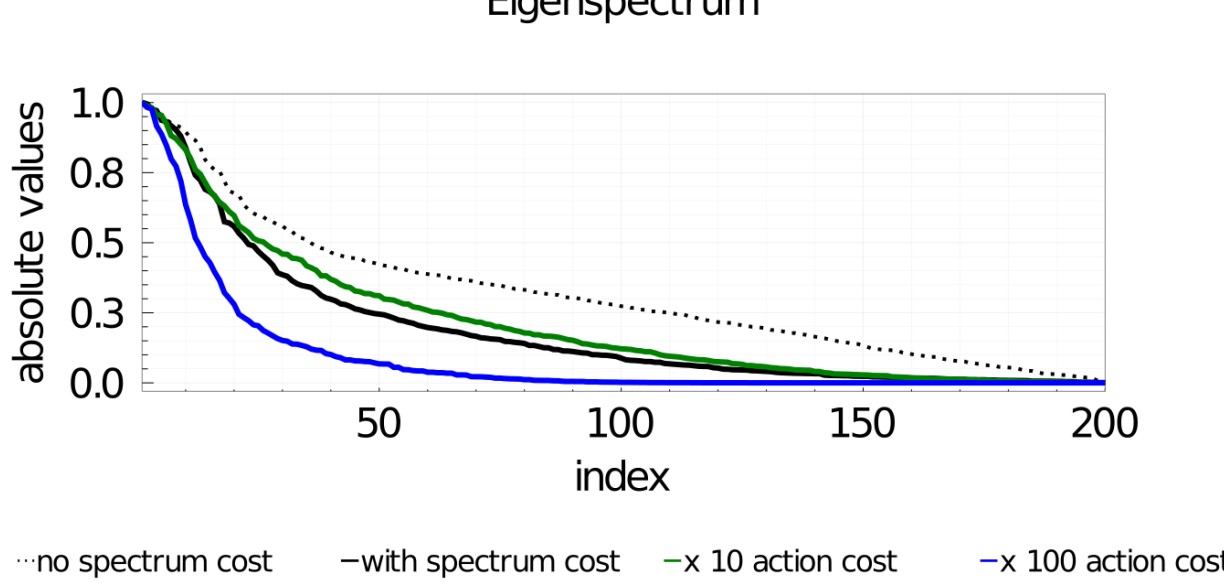

Figure 19: Averaged eigenspectrums showing absolute values of eigenvalues for the dynamics with/without the spectrum cost and with 10 times more action cost and 100 times more action cost.

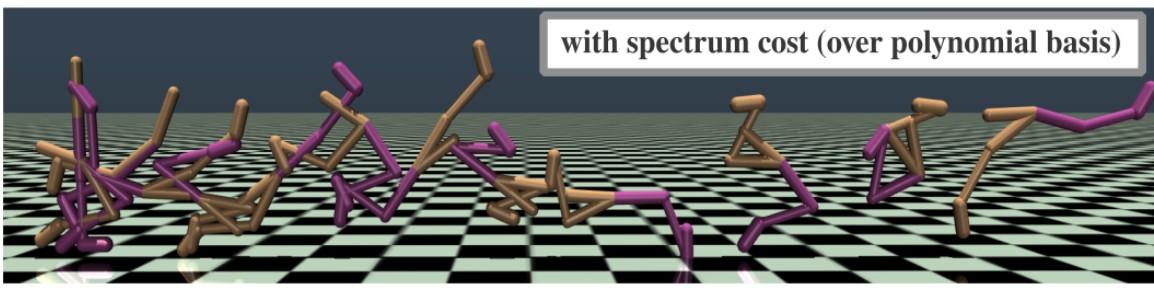

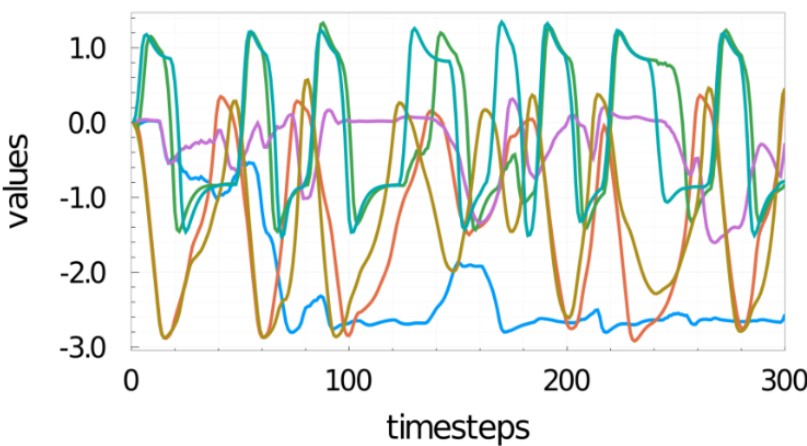

Figure 20: Up: Visualization of Walker motions generated by the CEM algorithm with the spectrum cost where the feature space is spanned by $x, x^2, x^3, x^4, x^5$. The motion is of that showing median value of the spectrum cost within the seed runs. Down: Its joint trajectory.

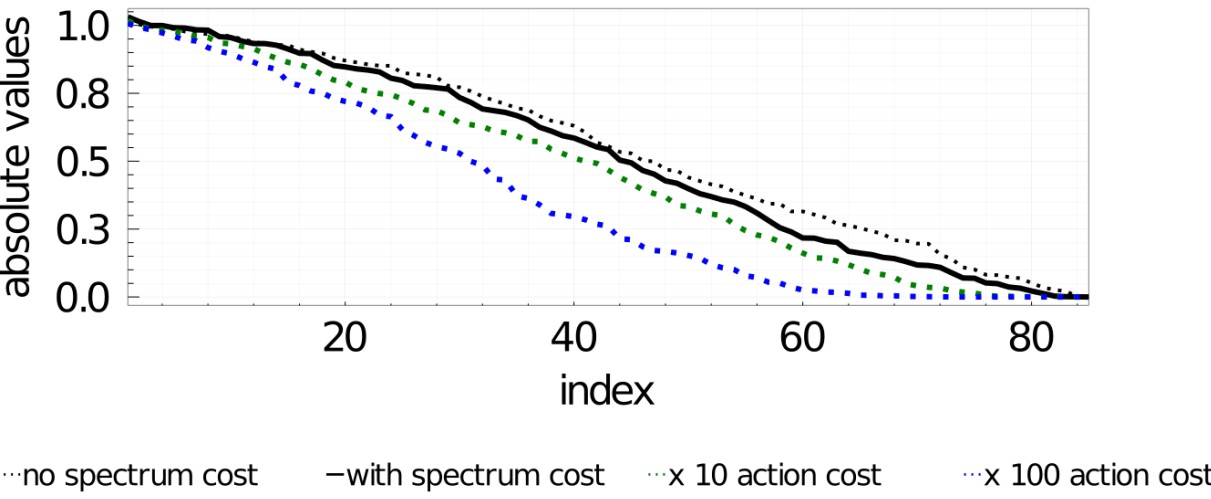

Figure 21: Averaged eigenspectrums over the polynomial feature space showing absolute values of eigenvalues for the dynamics with/without the spectrum cost and with 10 times more action cost and 100 times more action cost.

