# OpenReview forum: "Koopman Spectrum Nonlinear Regulators and Efficient Online Learning"
_TMLR — Accepted by TMLR_

### Review · Reviewer_62Mx · 2024-03-10

**Summary Of Contributions:**

The paper presents a method for control of nonlinear systems using Koopman operator theory with a spectral-based cost. The approach is claimed to be effective at online learning.

**Audience:**

Yes

**Broader Impact Concerns:**

The paper did not provide a broader impact statement.

**Claims And Evidence:**

No

**Requested Changes:**

The paper should consider more baseline experiments that holistically demonstrate and analyze the proposed method. For instance, adding in ablation studies regarding the basis function and choices of cost parameters would strengthen the contribution of the paper.

With regards to the results, the paper claims to have cyclical behavior for the cart pendulum and the walker, however, the results only show one "cycle". How repeatable are the trajectories only some of the results show clear repeatability. It would be good to see if the approach can be extended to non-cyclical systems. If not, what are the conditions for which a system can use the proposed method?

Additional results for the online learning for different systems and different basis functions would be valuable for the paper as well.

**Strengths And Weaknesses:**

The paper presents a novel form of using the Koopman operator which is a timely and useful tool for modeling nonlinear systems as linear systems in a lifted spectral domain. Control in that domain is often very challenging to acquire, even though the system is modeled as a lifted linear system. The work does a good job of describing the Koopman operator and how one can obtain controllers that leverage the spectrum of the nonlinear system. The theoretical results are very strong and seem to be a significant advancement in Koopman operator theory.

However, the empirical results of the proposed work are not convincing, especially with the baseline comparisons. For example, the provided video experiments do not seem like the proposed method provided much more improvement. The results provided in the paper provide a better picture of the results but are still limited in the example cases and could be explicitly compared with the addition of a table outlining the extent to which performance has improved.

---

> ### Author Response · Authors · 2024-04-04
> **Thank you for finding significant values in our findings.**
>
> We sincerely thank you for kindly taking your precious time reviewing our paper.
> We are happy that you find significant values in our theoretical findings and thank you for your comments.
> Given your comment, we added the broader impact statement section at the end while we believe our work does not carry a significant risk of harm.
> Also, we responded to your concerns below.
>
> >The paper should consider more baseline experiments that holistically demonstrate and analyze the proposed method. For instance, adding in ablation studies regarding the basis function and choices of cost parameters would strengthen the contribution of the paper.
>
> Thank you for the comment; given your comments we added several more experiments: (1) we added cartpole stabilization for half and double horizons; this is related to your comment below as well.  (2) we added more investigations into Walker experiments; we compared it to the case with increased action costs and measured them by another smoothness metric, and we added an experiment with different basis functions.  (3) linear example for the online learning algorithm to show its correctness (please also see our response to your last comment below).  We added those experiments in appendix except for (3) which is in the main body of the manuscript.
>
> At the same time, we would like to kindly mention that the purpose of our work is not to ''beat'' against the MDP counterpart, but to show some effects and intuitive understandings of our new framework, KSNR, from several point of views including theoretical analysis, simple simulation experiments and reduction to eigenstructure assignment problems.  We agree that it is a promising direction of research to further scale our methodology to match the maturity of the MDP counterpart as we described in the limitation section (since the study of MDPs has a long history); however, it is not a trivial work at all and we leave it for the future work.  We believe we are not making over-claiming statements in this respect.
>
> >Additional results for the online learning for different systems and different basis functions would be valuable for the paper as well.
>
> Thank you for the suggestion; in order to clearly convey the idea, although it is simple, we added linear system example.  Linear example basically corresponds to eigenstructure assignment problems but our work gives theoretical guarantees and we thought adding this experiment helps.  For this linear system that satisfies the assumptions, we confirm that our algorithm works correctly.  Note in this case the Koopman operator is realized over the canonical basis functions.
> However, as described in the limitation, we admit that this theoretical algorithm in its naive form may not scale well to the complicated control domains (we also added more sentences to the limitation section as well).

---

> ### Author Response · Authors · 2024-04-04
> **Additional responses**
>
> >With regards to the results, the paper claims to have cyclical behavior for the cart pendulum and the walker, however, the results only show one "cycle". How repeatable are the trajectories only some of the results show clear repeatability. It would be good to see if the approach can be extended to non-cyclical systems. If not, what are the conditions for which a system can use the proposed method?
>
> Thank you for the question; we think what you ask here is an important aspect, and we would like to clarify this better.  First, the purpose is not to let the agent demonstrate ''cycle'' motion, but to let the dynamics stable or smooth in the sense of the spectrum cost.  If we are aiming to create a cyclic motion, it would be the case that we can achieve this solely by cumulative cost; however it is a post hoc design of the motion.  Our focus is to effectively and systematically regularize or constrain the dynamics within the spectrum domain.
>
> To clarify this more, please imagine Fourier analysis as an analogy (although ours is not the duality in the same sense; we added an illustration after introducing the framework); once we know the certain target time sequence (or wave), it is possible to describe this in the time domain.  However, if we just wish to do low-pass filtering, or some manipulation or restriction over its frequency domain, it is unclear what kind of time sequence should be obtained until such operations in the frequency domain is done.  In our framework KSNR, what we aim for is to add regularization of Koopman spectrum domain to systematically obtain certain dynamics having desired global property (such as stability) while the cumulative cost acts as an analogy to the time domain in Fourier analysis.  We added this explanation with an illustrative picture in Section 3.2 as well.
>
> To strengthen this point, we also added cartpole experiments with different time length (50 and 200).  While for the case of 200, the CEM planner could not find a policy that keeps the pole straight up, we see that the longer horizon may not indicate more oscillation ''cycles''.  In fact, our spectrum cost only regularizes the dynamics to be stable, which may include a motion that shows ''convergence'' of the speed to a fixed constant, which is stable over the state that excludes position information.  Note, without spectrum cost, it shows ever-increasing speed.
>
> We are aware that our CEM heuristic has limitation in the sense that it does not necessarily find a global optimal solution, and please note we are not claiming that the presented motion is the global optimal solution with respect to our cost.
> Lastly, we mention that the Walker example is not one ''cycle'', but we observe that it tends to fall over at the last moment.  One reason would be because of the limitation of CEM heuristic for longer time horizon.
> To recap, KSNR is not only for the cyclic motion, but is more broadly for the spectrum regularization in the Koopman spectrum domain.
>
> Finally, not only a limit cycle motion but a convergent behavior also corresponds to a Koopman mode as well; therefore our framework naturally embraces non-cyclic motions as target.  Hope this answers to your comment.

---

> > ### Comment · Reviewer_62Mx · 2024-04-10
> > **Response to comments**
> >
> > Thank you for the response. This clarifies a few things; however, if the point want to produce a stable Koopman operator model, work like this does exist:
> >
> > Mamakoukas, Giorgos, Ian Abraham, and Todd D. Murphey. "Learning stable models for prediction and control." IEEE Transactions on Robotics (2023).
> >
> > which learns a Koopman operator which has spectral modes restricted to the unit norm. Would not just keeping the Koopman operator within a stable norm yield a similar result?
> >
> > Regarding the cartpole, would it be possible to show what happens the pole angle for a longer time duration? The paper does not have this result (just the position which is not usually the focus point of the cart pole balance problem). The attached multimedia also does not seem like the pole is actually stable (or approaching being stabilizable).

---

> ### Author Response · Authors · 2024-04-12
> **Thank you so much for reading our response**
>
> Thank you so much for reading our response and for your important questions which in fact are critical for understanding the core idea of our work; below we answered to your questions, and revised our manuscript accordingly.
>
> >if the point want to produce a stable Koopman operator model, work like this does exist: (Mamakoukas, Giorgos et al) which learns a Koopman operator which has spectral modes restricted to the unit norm. Would not just keeping the Koopman operator within a stable norm yield a similar result?
>
> Thank you for asking about a question that will help further clarify the core idea of our work.  Our focus is *not to learn the stable Koopman operator* but to solve the decision making problem that balances the (possibly task-based) cumulative cost and the spectrum cost that could be used to enforce stability.  On the other hand (thank you for introducing the work), the goal of the work in (Mamakoukas, Giorgos et) is to learn the stable Koopman operator realized over properly chosen feature space for a given dynamics which may be unstable, and they use it to predict the long-term behaviors or to construct a (control) Lyapunov function for example.
>
> We think answering to this question is very important to avoid possible but critical misunderstandings so we provide a careful response below.
>
> First of all, what we need to obtain is the spectrum information or cost of the Koopman operator realized over a chosen space for a given dynamics parameter $\Theta$, and it should reflect the information such as unstability as well, for the sole purpose of decision making, i.e., identify $\Theta$ that solves the problem (3.2).  Therefore, please note we are not aiming to obtain a closest stable matrix to the true Koopman operator.
> Of course you may obtain a (control) Lyapunov function from the stable Koopman operator when we already know a nearly stabilizing control policy (where the error margin is sufficiently small).  However, (1) it is a post-hoc design after you have this nearly stabilizing control policy, and (2) the obtained control Lyapunov function guarantees stability under certain conditions but adds too restrictive condition.  For example, consider a very simple single integrator dynamics.  Suppose we obtain a control Lyapunov function which guarantees the stabilization of the system towards the zero state (or certain set of states).  This function can be used in a downstream task *only to guarantee stability towards the zero state or to guarantee the invariance of its level sets etc.*, which means the constraint with this Lyapunov function will produce a strictly smaller set of dynamics than the entire set of stable dynamics which admit different types of Lyapunov functions.
>
> To recap, our purpose is to identify a good parameter $\Theta$ that will produce the dynamics having desirable balance of cumulative cost and spectrum cost as a decision making problem and not to learn a stable Koopman operator or a control Lyapounov or barrier function.
>
> As a short answer to your question, keeping the Koopman operator within a stable norm *does not* yield the same result.  We assume your question is about the resulting behaviors of KSNR applied to stability regularization and of the one generated after one constructs a control Lyapunov function based on the learned stable Koopman operator that is the closest to the true operator.  For example, if the stable Koopman operator is learned by a trajectory that stabilizes the cartpole around the zero position and velocity to keep the pole straight up, the downstream control actions that are constrained by a control Lyapunov function obtained by this stable Koopman operator all stabilize the cartpole in this way.  However, stable dynamics include other behaviors such as oscillating motions or convergence to certain position other than the zero states.  If your question is about the property of the Koopman operator itself, then please refer to the above response, i.e., our focus is *not to learn the stable Koopman operator* but to solve the decision making problem that balances the (possibly task-based) cumulative cost and the spectrum cost that could be used to enforce stability.
>
> To clarify this argument, we added in the related work section and in the simulation part some remarks and an illustration portraying several stable attractors that are included in the set of stable dynamics.  Also, we cited the paper you have introduced, so please also refer to the manuscript.
>
> Finally, while irrelevant to this question, we also mention that, for control problems, a nonlinear dynamics cannot be described by the features that depend only on the state or on action in general.  Note our model subsumes such a very specific case as well.

---

> > ### Author Response · Authors · 2024-04-12
> > **Additional response**
> >
> > >Regarding the cartpole, would it be possible to show what happens the pole angle for a longer time duration? The paper does not have this result (just the position which is not usually the focus point of the cart pole balance problem). The attached multimedia also does not seem like the pole is actually stable (or approaching being stabilizable)
> >
> > Thank you for the comment; we added the plots of the pole angle in the appendix as well.  As we noted in the manuscript, we emphasize that for the case of time horizon $200$, the CEM planner could not find a solution that keeps the pole up right (although it should exist), meaning the cumulative cost suffers from this failure while the spectrum cost still shows the separation between the cases with / without the spectrum cost regularization.
> >
> > Please note that the stability here does not mean the stability towards the zero position / velocity or stability of the pole being straight up position.   We think this would be the source of confusion; our stability regularization aims at restricting the dynamics to the set of stable dynamics (although this is not perfectly encoded as we use finite dimensional approximation).  Therefore, in this cartpole example *over the state space where the angle representation is bounded*, it is to find the dynamics that gain as much velocity as possible within the set of stable dynamics which includes the case of failure of keeping the pole straight up; while the preference of the pole being straight up is encoded by the cumulative cost.  In other words, failure of keeping the pole straight up itself does not directly violate the stability constraint but increases the cumulative cost, and how much penalty to assign this violation is a design problem.
> >
> > And in our examples, while all of the horizon cases anyway show well-regularized spectrum costs, the case of horizon $100$ shows particularly interesting motion.
> >
> > If we want to regularize the dynamics so that it satisfies the stability towards the zero state, then one should indeed use control Lyapunov functions or possibly the different state representation.  Please note however that the enforcement of Lyapunov constraints every step can be encoded by the cumulative cost, and it is anyway within this broad decision making framework.
> >
> > To further clarify this argument, we extended the remark in the simulation section as well.

---

### Review · Reviewer_KaJJ · 2024-03-20

**Summary Of Contributions:**

This paper studies the problem of controlling a nonlinear system by minimizing both stepwise costs and also a koopman spectrum cost. The main intuition is tha this serves as a regularizing effect that leads to smoother motions that are more natural and adapted to the dynamics of the system itself. Both simulation experiments and some theory for an online learning algorithm are demonstrated.

**Audience:**

Yes

**Claims And Evidence:**

Yes

**Requested Changes:**

I only have requests for clarifications based on these questions:

-  “Note, this assumption does not preclude the necessity of exploration…” Can you elaborate on this point? Is the exploration that is done in the algorithm done through maintaining the BALL confidence set and minimizing over this optimistically? Then, what purpose does this eigenvalue lower bound serve in Assumption 5?
- In the initial simulation experiments of section 3.3,  it’s a little unclear what is being learned vs. just optimized. The appendix seems to suggest that the K is being learned through transitions. How were these generated? Was this done iteratively? If so, I’m confused about the difference between these and the results 4.4. Why can one not use the method of 3.3 to solve 4.4 as well?
- Is it possible to achieve the desired smoothness effects by simply regularizing e.g. punishing high accelerations? Qualitatively, how would this differ from the proposed method? It would perhaps be nice to see comparisons with this in the experiments.
- I don’t really understand the design decisions made in the first cart pole experiments. The reward is set high for the cart for large velocity, so it makes sense that it increases the velocity. If one wanted it to oscillate, why not just shape the reward function to get it to oscillate? It’s unclear why this might require more machinery when the spectrum cost is also set by hand.
- The regret bound seems potentially dependent on the adversary’s choice of sequences, as opposed to doing regret that is just the suboptimality with respect to the lowest cost achievable by any control policy. Can you elaborate on how the sequences are chosen? Are they not chosen by the learner’s policy itself?

**Strengths And Weaknesses:**

Strengths:
- The paper addresses an important problem of learning/optimizing control policies that lead to smooth, natural motion in contrast to what might be learned by an RL algorithm
- The results are well-grounded in theory to motivate the design decisions.
- Simulations seem to suggest this achieves the desired results.
- The theory for the online learning algorithm is also promising.

Weaknesses:
- a number of approximations have to be made to get the online algorithm to work on cartpole, which is a relatively simple problem (including restricting the policy space). This suggests it may be difficult to scale these methods and insights to harder control problems.
- I have questions about some of the experiments below that are potential weaknesses.

---

> ### Author Response · Authors · 2024-04-04
> **Thank you so much for your important questions.**
>
> We sincerely thank you for kindly taking your precious time reviewing our paper.
> We thank you for finding values in our work and also for very constructive comments.
> Below, we answered to your questions and we revised the manuscript accordingly.
>
> >a number of approximations have to be made to get the online algorithm to work on cartpole, which is a relatively simple problem (including restricting the policy space). This suggests it may be difficult to scale these methods and insights to harder control problems.
>
> Thank you for pointing it out; as we described in the limitation section, it is true that our learning algorithm requires a number of assumptions.  It may be possible that there exists a clever way of incorporating feature learning etc. together to efficiently scale up; since it is not a trivial addition, we leave it as a future work, but we added slightly more sentences to the limitation section.
>
> >''Note, this assumption does not preclude the necessity of exploration…'' Can you elaborate on this point? Is the exploration that is done in the algorithm done through maintaining the BALL confidence set and minimizing over this optimistically? Then, what purpose does this eigenvalue lower bound serve in Assumption 5?
>
> Thank you for the comment; yes the exploration occurs through optimism in the face of uncertainty.  Assumption 5 intuitively states that fitting the transition data over the feature space is not ill-posed.  If the sole purpose is just to fit the data for single policy parameter, we may not need a well-designed exploration strategy in this case; however, since the eigenvalues of $\sum_{n=0}^{N^t-1}\sum_{h=0}^{H^t_n-1}{\mathcal{A}^t_{h,n}}^\dagger \mathcal{A}^t_{h,n}$ is not bounded below by a positive constant, we still need to design an exploration strategy carefully.  As such, our problem cannot be solved straightforwardly and KS-LC$^3$ uses optimism in the face of uncertainty tailored for our problem.
>
> Given your comment, we added more words in the remark.
>
> >In the initial simulation experiments of section 3.3, it’s a little unclear what is being learned vs. just optimized. The appendix seems to suggest that the K is being learned through transitions. How were these generated? Was this done iteratively? If so, I’m confused about the difference between these and the results 4.4. Why can one not use the method of 3.3 to solve 4.4 as well?
>
> Thank you for the question; we added the clarification sentence in the relevant part.  For 3.3, we assume to have access to a nominal simulator to plan as like classical MPC algorithm while we use the learned environment for planning in 4.4 (as like model-based RL).  For reference, the added sentence is the following: Specifically, at each iteration of CEM, we generate many parameters ($\Theta$s) to compute the loss (i.e., the sum of the Koopman spectrum cost and *negative* cumulative reward).  This is achieved by fitting the transition data to the chosen feature space to estimate its (finite-dimensional realization of) Koopman operator (see Appendix H.1); here the data are generated by the simulator which we assume to have access to.
>
> >Is it possible to achieve the desired smoothness effects by simply regularizing e.g. punishing high accelerations? Qualitatively, how would this differ from the proposed method? It would perhaps be nice to see comparisons with this in the experiments.
>
> Thank you for an insightful comment; to answer to your question, we conducted more experiments on the Walker environment and added the section in appendix.  We tested x10 action cost case and x100 action cost case, and also measured the smoothness by the average absolute change of joint positions in addition to the spectrum cost itself.  In short, increased action cost in our scenarios shows lower spectrum cost; while KSNR shows better cumulative reward with better spectrum cost.  However, the motion generated by the increased action cost shows lower action penalty cost; which implies that the spectrum cost and the action cost have some correlation while they qualitatively prefer different motions.
> The smoothness measure seems consistent to the spectrum cost in this case.
> Also, we tested with different feature space, namely, the polynomial basis, and for this case, $10$ times more action cost led to lower spectrum cost than KSNR while the cumulative reward of KSNR becomes higher than the case with RFF feature.  Qualitatively, KSNR now shows similar rolling behavior to the motion without the spectrum cost but with slightly better periodicity.  These results imply that the feature space selection influences the qualitative behavior difference in practice; please also see the Remark added after Example 3.1, which basically states that the feature space selection is an important part of the KSNR framework.

---

> ### Author Response · Authors · 2024-04-04
> **Additional responses**
>
> >I don’t really understand the design decisions made in the first cart pole experiments. The reward is set high for the cart for large velocity, so it makes sense that it increases the velocity. If one wanted it to oscillate, why not just shape the reward function to get it to oscillate? It’s unclear why this might require more machinery when the spectrum cost is also set by hand.
>
> Thank you for your question; our description in the original manuscript might have been misleading.  We are not trying to force the cartpole to oscillate, but rather to let the dynamics be stable in the sense that the Koopman operator over a chosen space has less than or equal to one spectral radius.  As you kindly point out, it may be possible to carefully design an immediate cost function to let the cartpole oscillate, but we mention that this is a post hoc design.  To clarify this more, please imagine Fourier analysis as an analogy (although ours is not the duality in the same sense); once we know the certain target time sequence (or wave), it is possible to describe this in the time domain.  However, if we just wish to do low-pass filtering, or some manipulation or restriction over its frequency domain, it is unclear what kind of time sequence should be obtained until such operations in the frequency domain is done.  In our framework KSNR, what we aim for is to add regularization of Koopman spectrum domain to systematically obtain certain dynamics having desired global property (such as stability) while the cumulative cost acts as an analogy to the time domain in Fourier analysis.  We added this explanation with an illustrative picture in Section 3.2 as well.
> To strengthen this point, we also added cartpole experiments with different time length (50 and 200) in the last part of the appendix.  While for the case of 200, the CEM planner could not find a policy that keeps the pole straight up, we see that the longer horizon may not indicate more cyclic motions.  In fact, our spectrum cost only regularizes the dynamics to be stable, which may include a motion that shows ''convergence'' of the speed to a fixed constant, which is stable over the state that excludes position information.  Note, without spectrum cost, it shows ever-increasing speed.
> To recap, this experiment is not to show that the spectrum cost makes the cartpole oscillate necessarily, but to show that it adds stability constraint effectively and systematically.
>
> In the manuscript, we added the illustration and explanation about this point, i.e, an analogy to the Fourier analysis.  Also, we added a note in the text corresponding to the cartpole simulation.
>
> >The regret bound seems potentially dependent on the adversary’s choice of sequences, as opposed to doing regret that is just the suboptimality with respect to the lowest cost achievable by any control policy. Can you elaborate on how the sequences are chosen? Are they not chosen by the learner’s policy itself?
>
> Thank you for your question; we think this point is important and we would like to clarify better.  First of all, the adversary only chooses the initial states, their time horizons, and the immediate cost, but the trajectories themselves are generated by the learner's policy.  Therefore, if the given assumptions hold, our regret bound holds no matter what initial states or horizons the adversary chooses.  However, a caveat here is that to satisfy Assumption 5 in practice, the number $N^t$ might become larger and it would make the bounds $H$ (in Assumption 5) and $V_{\rm max}$ large.  Also, to satisfy Assumption 6 under a possibly large scale immediate cost function, $V_{\rm max}$ can be made larger.  In such cases, because the regret bound depends on $H$ and $V_{\rm max}$, the sample complexity may get worse.  Please note however that the theorem is valid anyway.  To clarify this, we added remark after Theorem 4.6.

---

> > ### Comment · Reviewer_KaJJ · 2024-04-10
> > **Response**
> >
> > Thank you for detailed responses. This is very informative. A follow up question on Assumption 5: I think I understand the role of optimism more now, but I'm still confused about why this is assumption is necessary at all. Can you point out the equation / problem that would be ill-posed without this? Why can one not just add regularization, which is commonly done optimistic algorithms? Also, C does appear in the bounds of the theorem, but somehow the text suggests that you can make C larger by generating more trajectories? I'm just trying to get a sense of what role it plays.

---

> > > ### Author Response · Authors · 2024-04-12
> > > **Thank you so much for reading our response**
> > >
> > > Thank you so much for reading our response and for your important followup questions; below we answered to your questions, and revised our manuscript accordingly.
> > >
> > > >A follow up question on Assumption 5: I think I understand the role of optimism more now, but I'm still confused about why this is assumption is necessary at all. Can you point out the equation / problem that would be ill-posed without this? Why can one not just add regularization, which is commonly done optimistic algorithms?
> > >
> > > Thank you for the question; as a high-level overview, it is required to simultaneously manage the cumulative cost and the spectrum cost through the use of common confidence balls; note the former cares about each single-step transition while the latter deals with the global dynamics represented as the Koopman operator realized over a given space.  And we need to show that the optimism in the face of uncertainty through the use of the common confidence balls optimizes the trade-off between exploration and exploitation for the sum of two costs that are distinct.
> > >
> > > The direct use of this assumption is highlighted by Lemma F.2 (derived from our positive operator norm bounding lemma) which is used to basically bound the norm of difference between the true Koopman operator and the estimated one at each episode by the multiple of *radius* of the confidence ball for the term1 in (F.1).  Actually, $\Sigma_{\mathscr{A}}^t$ is indeed defined by using the regularization $\lambda I$; also we defined $\Sigma_{\mathscr{B}}^t$ to exploit in the analysis and it also has this regularization term as well.
> > >
> > > If we consider including regularization terms to $\mathbf{\phi}\mathbf{\phi}^\dagger$ to artificially satisfy Assumption 5, on the other hand, we end up needing this term *every episode*, and it will cause linear regret due to this *fake transition data*.
> > >
> > > However, this argument that relates the cumulative cost and the spectrum cost to manage them through the use of common confidence ball is one of the critical procedures and it might be the case that there might exist other ways of managing them with a different set of assumptions and with different algorithms.  And we thank you for helping us clarify the arguments; we extended the remark for this assumption in the manuscript as well.
> > >
> > > >Also, C does appear in the bounds of the theorem, but somehow the text suggests that you can make $C$ larger by generating more trajectories? I'm just trying to get a sense of what role it plays.
> > >
> > > Thank you for the comment; as you mention, the regret bound depends on $C$, and it could be made larger when there are more trajectories every episode in case they contribute effectively to increase the smallest eigenvalue.  However, the number of trajectories is determined by the adversary which at least is assumed to satisfy Assumption 5, and we assume we somehow only know this given $C$.  In addition, even when the adversary generates more trajectories, it would also increase $H$ and $V_{\rm max}$; as such the preference of the number of trajectories (even if the number is controllable) is not straightforward.
> > > We hope this answers to your question.

---

### Review · Reviewer_qbzi · 2024-03-24

**Summary Of Contributions:**

This manuscript proposes a new paradigm for learning control of nonlinear random dynamical systems, which is based on Koopman operator theory. Specifically, the authors propose an optimization formulation over the cost of the Koopman operator of the dynamical system. They term this the Koopman spectrum cost.  Contributions of the manuscript include the conceptual formulation and proposal, a (theoretical) learning algorithms, as well as several numerical examples illustrating the proposed method.

**Audience:**

Yes

**Broader Impact Concerns:**

No concerns.  The manuscript contains a description of (technical) limitations, which is commendable.

**Claims And Evidence:**

Yes

**Requested Changes:**

See weaknesses above.

Minor comments can be considered optional suggestions.

**Strengths And Weaknesses:**

While this reviewer is not an expert on Koopman theory for dynamical systems, they get the impression that this manuscript makes a solid and relevant contribution to the theory of dynamical systems / controller learning in the context of Koopman operator theory.



### Strengths

1) Manuscript makes a conceptual and potentially fundamental contribution, which could open up new research directions (as also mentioned in the manuscript)
2) Well written manuscript
3) Clear and transparent communication of contributions (p. 2)
4) Assumptions clearly stated
5) Aiming to give illustrative examples and explanations



### Weaknesses

1) While the authors make an effort providing intuition for the theoretical concepts, especially linking the exposition to the (more commonly known) MDP-type RL framework, which is commendable, the exposition could be still be clearer or more elaborate in parts. This includes, in particular, the following parts:
   * P. 4, "is a mapping that takes a Koopman operator as an input and returns its cost" -> Can we interpret this to basically mean that the mapping takes a dynamical system (described by a Koopman operator) as input and returns its cost?  Such statement could be added and also compared to standard MDP formulation for illustration.
   * What is the goal of the design in (3.2)? In practice, would this be used to find a controller? The controller is not explicitly introduced, but presumable it is considered part of the dynamical system (thus its parameters would be part of \Theta). I suggest to clarify this. In this context, what components in the formulation have the role of the controller/policy?
   * Sec. 3.3: How is CEM based policy search related to the design problem (3.2)?  Is CEM the optimizer that is used to solve the problem (3.2)?  Or is this a separate problem building on top of it?  In general, it would be very helpful if the authors could better explain how policy search relates to their design problem (3.2).
2) If possible, it would be helpful to provide intuition about Assumptions 2-4 + 6 (e.g., examples for which the assumptions hold; are the assumptions restrictive; etc)
3) Some further aspects still require more explanation (see minor comments and suggestions below)



### Minor comments and suggestions

* Introduction/Abstract: What do the authors mean with "unpredictability" of the motions generated by standard RL? Please explain.
* p. 2, key takeaways: I don't understand the sentence "such, we need some specific structural assumptions that are irrelevant to the Kernelized Nonlinear Regulator (KNR)". Why is KNR mentioned here? It seems to be out of context, or at least the context is not clear to the reviewer.
* Related work: "Skill learning", e.g. with RL, is an important topic in robotics.  I suggest the authors to review main works there and mention the relation to their approach herein.  Some of the main authors include Jan Peters, Jens Kober, Stefan Schaal, Freek Stulp, for example.
* Section 2: At the end of this section, "population based policy search" is mentioned, but not explained.  It was unclear to this reviewer what the authors mean. Furthermore, why are these numerical examples "based on population based policy search" relevant for this work?  Please explain.
* Bottom of p. 3: even though it is standard, "i.i.d." should be introduced.
* General question:  Does "Regulator" refer to "regularization" or "Regulator" (controller) as in LQR?  Might be useful to clarify early on. (I only realized later, what was actually meant.)
* Section 4, first paragraph: What is meant by "model-based algorithm"?  What model?

---

> ### Author Response · Authors · 2024-04-04
> **Thank you so much for your very constructive comments.**
>
> We sincerely thank you for kindly taking your precious time reviewing our paper.
> Also, we thank you for very constructive questions which have helped us improve the clarity of some of the contents.
> We are happy that you value our ideas.
> Below, we answered to your questions and we revised the manuscript accordingly.
>
> >P. 4, "is a mapping that takes a Koopman operator as an input and returns its cost" Can we interpret this to basically mean that the mapping takes a dynamical system (described by a Koopman operator) as input and returns its cost? Such statement could be added and also compared to standard MDP formulation for illustration.
>
> Thank you for the question; the answer depends on the space $\mathcal{H}$ to take in general.  If the Koopman operator over $\mathcal{H}$ *fully represents* the dynamics in the sense that it can reproduce the dynamical system over the state space, the spectrum cost is viewed as a cost that directly takes the dynamical system itself as an input.  On the other hand, depending on the choice of $\mathcal{H}$, it is often the case that the Koopman operator does not uniquely reproduce the dynamics (e.g., the extreme case is a one-dimension space spanned by a constant function).  For such cases, the cost only acts as a regularizer.  Compared to the MDP formulation, which aims at representing the dynamics by the cumulative cost incurred on local trajectories, which resembles the principle of least action, our KSNR considers global property of the dynamics (i.e., spectrum).  We also added some explanations with an analogy to Fourier analysis in Section 3 which clarifies this point as well.
>
> >What is the goal of the design in (3.2)? In practice, would this be used to find a controller? The controller is not explicitly introduced, but presumable it is considered part of the dynamical system (thus its parameters would be part of $\Theta$). I suggest to clarify this. In this context, what components in the formulation have the role of the controller/policy?
>
> Thank you for the question; yes it is used to find a dynamics generated by a parameter $\Theta$.  In control setting, $\Theta$ is indeed a controller, and determines the cost $\Lambda$ (spectrum cost) and the cumulative cost $J$ (typical MDP cost). Based on your suggestion, we added the clarification after introducing (3.2).  We state the sentence here for reference: In control problem setting, the problem (3.2) is to find a *control policy* $\Theta^*$, which minimizes the cost; note, each control policy $\Theta$ defines the resulting dynamical system that gives the costs $\Lambda$ and $J$ in this case.  However, we mention that the parameter $\Theta$ can be the physics parameters used to design the robot body for automated fabrication or any parameter that uniquely determines the dynamics.
>
> >Sec. 3.3: How is CEM based policy search related to the design problem (3.2)? Is CEM the optimizer that is used to solve the problem (3.2)? Or is this a separate problem building on top of it? In general, it would be very helpful if the authors could better explain how policy search relates to their design problem (3.2).
>
> Thank you for the question; CEM is one of the heuristic approach that can be used to solve (3.2).  Based on your suggestion, we added clarification when introducing CEM approach.
>
> >If possible, it would be helpful to provide intuition about Assumptions 2-4 + 6 (e.g., examples for which the assumptions hold; are the assumptions restrictive; etc)
>
> Thank you for the question; we added remarks after Assumptions 2-4 + 6 as well with intuitive descriptions and their restrictiveness.
>
> >Introduction/Abstract: What do the authors mean with ''unpredictability'' of the motions generated by standard RL? Please explain.
>
> Thank you for pointing it out; we added the following sentence after the relevant part in the introduction: Intuitively, the motion specified by the task-oriented cumulative cost formulation may ignore ''how to'' achieve the task unless careful design of cumulative cost is in place, necessitating a systematic approach that effectively regularizes or *constrains* the dynamics to guarantee predictable global property such as stability.
>
> In short, our KSNR framework can systematically let the dynamics have stability etc. that gives predictability of the global property of the dynamics at least.  Specifying such dynamics characteristics by cumulative cost formulation is often harder as we described in Section 3.2 for example (please also refer to the added explanations with an analogy to Fourier analysis).

---

> ### Author Response · Authors · 2024-04-04
> **Additional responses**
>
> >p. 2, key takeaways: I don't understand the sentence "such, we need some specific structural assumptions that are irrelevant to the Kernelized Nonlinear Regulator (KNR)". Why is KNR mentioned here? It seems to be out of context, or at least the context is not clear to the reviewer.
>
> Thank you for the question; it is because we use several theoretical techniques developed for KNR, and we believe we should mention this and explain how our Koopman formulation requires additional theoretical arguments which are in fact essential to consider non-MDP formulation.  Given your question, we added a clarification in the subsection of "Relations to the kernelized nonlinear regulator and to eigenstructure assignments" in the manuscript.
>
> >Related work: "Skill learning", e.g. with RL, is an important topic in robotics. I suggest the authors to review main works there and mention the relation to their approach herein. Some of the main authors include Jan Peters, Jens Kober, Stefan Schaal, Freek Stulp, for example.
>
> Thank you for the suggestion and for the introduction of work; our framework indeed has relation to the work on motor primitive as well.  It considers parametric nonlinear dynamics having some desirable properties such as stability, convergence to certain attractor etc., and it is related to the Koopman spectrum regularization in the sense both aim at regulating the global dynamical properties.  For more broad aspect, KSNR is related to skill learning in a bit abstract sense that our KSNR framework tries to consider ''HOW'' aspect of the behavior generation in addition to the task-based ''WHAT'' aspect (please also see the response above about ''unpredictability'').  The notion of skill is also orthogonal to the task, and finding a task-executing behavior within a sequence of (learned) skills or within a parameterized skill space can have regularizing effects.  Given your comments, we added the following texts to the related work section citing some of the work from the authors you mentioned: Historically, the motor primitives investigated in, for example, (Peters \& Schaal, 2008; Ijspeert et al., 2002;
> Stulp \& Sigaud, 2013) have considered parametric nonlinear dynamics having some desirable properties such as stability, convergence to certain attractor etc., and it is related to the Koopman spectrum regularization in the sense both aim at regulating the global dynamical properties.
> Those primitives may be discovered by clustering (cf. Stulp et al. (2014)), learned by imitation learning (cf. Kober \& Peters (2010)), and coupled with meta-parameter learning (e.g., Kober et al. (2012)).
>
> >Section 2: At the end of this section, "population based policy search" is mentioned, but not explained. It was unclear to this reviewer what the authors mean. Furthermore, why are these numerical examples "based on population based policy search" relevant for this work? Please explain.
>
> Thank you for the question; we added clarification to the manuscript.  We added the following sentence:  ... illustrative numerical examples based on population based policy search (e.g., genetic algorithm), followed by an introduction of its example online learning algorithm (Section 4) with its theoretical insights on sample complexity and on reduction of the model to that of eigenstructure assignments problem as a special case.  For more details about population based search that repeatedly evaluates the sampled actions to update the sampling distribution of action sequence so that the agent can achieve lower cost, see for example (Beheshti \& Shamsuddin, 2013).
>
> CEM that we use in this work to heuristically solve KSNR problem is one of the examples, and we added this clarification briefly as well after introducing CEM.
>
> >General question: Does "Regulator" refer to "regularization" or "Regulator" (controller) as in LQR? Might be useful to clarify early on. (I only realized later, what was actually meant.)
>
> Thank you for the question; we use "Regulator" as controller as in LQR in a general sense that it regulates the dynamics to have desirable characteristics.  As you mention, we use spectrum cost to regularize the dynamics, and it ''regulates'' the dynamics to have specific characteristics, and we believe both meanings do not contradict.  Given your question, we added some clarification there as well (we added the following sentence: Note that ``Regulator'' in KSNR means not only *controller* in control problems but a more broad sense of regularization of dynamical systems to obtain specific characteristics.)
>
> >Section 4, first paragraph: What is meant by "model-based algorithm"? What model?
>
> Thank you for the question; we are not directly learning the spectrum cost itself but the Koopman operator model, and in this sense we call this model-based.  To clarify this, we added the sentence: Here, ''model-based'' simply means we are not directly learning the spectrum cost itself but the Koopman operator model.

---

> > ### Comment · Reviewer_qbzi · 2024-04-26
> > **Reply to authors' response**
> >
> > Thank you for your responses to my questions and comments. These make sense to me and clarify the questions that I had.

---

> > > ### Author Response · Authors · 2024-04-27
> > > **Thank you so much for reading our responses**
> > >
> > > Thank you so much for reading our responses; and thank you for helping us improve the manuscript.

---

### Decision · Action_Editor_HWAr · 2024-05-20

**Recommendation:** Accept with minor revision

**Comment:**

I would like to thank the authors for their patience, as this paper has been under review for longer than usual.

The final decisions of the reviewers have been one Accept, one Leaning Accept, and one Leaning Reject.

All reviewers believe that the paper makes interesting contributions. So, from the novelty perspective, this paper is strong. Most of them also believe that the claims are reasonably well supported. The reviewer on the negative side, however, criticizes the paper because the advantages of the proposed method is not very evident.


I read the paper myself, without verifying the proofs. At a high-level, the paper has
- some conceptual novelty (the addition of a Koopman spectrum cost to the per-step cost);
- theoretical guarantees in the form of regret bounds (which are under strong assumptions, such as Assumption 1);
- some empirical illustrations, mostly on toy problems.

Given that the formulation is new and optimization of the corresponding objective is not trivial, perhaps it is reasonable not to expect extensive and competitive empirical results for this work.


One aspect of the paper that can and should be improved is its clarity. At several points in the paper, its quality of writing and clarity decline. I believe it is partly because the authors revised the paper to answer specific questions of the reviewers, but those revisions disrupted the flow of the paper (for example, Figure 3 and Analogy to Fourier analysis). In some other places, more explanation can be added. I provide some specific examples, in addition to some of my own questions.

**Overall, this is a good paper overall, and can be accepted if the authors improve the paper's clarity.**


===

Some places were clarity can be improved and some additional questions:

- More discussion of the RDS model in Section 3.1. For example, it is stated that "RDS subsume many practical systems including Markov chains". Some examples of this would be helpful.
- More explanation of the Koopman operator. The mathematical definition is in Definition 3.1, but some intuition would be helpful. For example, it is not clear whether such an operator K always exists or not.
- Example 3.1 can be expanded. How easy is it to compute the spectral radius of the Koopman operator (given that it is infinite dimensional)? What are the practical ways to compute it?
- Remark 3.1 is unclear to me. This is apparently an answer to one of the reviewers, but it may not be very clear for someone who hasn't read the discussion between the authors and the reviewers.
- m and m* are used in Section 3.3 before being introduced much later in Appendix H.2. In general, I believe the discussion of Simulated experiments in Section 3.3 can be improved by adding more details.
- Would you confirm that the minimum in the definition of regret is after the summation over t and not before that?

**Audience:**

This paper would be of interest to the TMLR audience, especially those interested in reinforcement learning, optimal control, and learning of dynamical systems.

**Claims And Evidence:**

Mostly yes.
On the theory side, the paper provides good evidence, under certain and perhaps strong assumptions.
On the empirical side, it provides some evidence about the method, but there has been concerns raised by one of the reviewers.

---

> ### Author Response · Authors · 2024-05-21
> **Thank you very much for your handling our paper**
>
> Thank you very much for your commitment to the process of our submission and to the TMLR community.  And thank you so much for your time reading our paper.
> We will improve the clarity of some places in the paper following your suggestions, and will answer to the questions by the specified time (hopefully earlier).
>
> Sincerely,
> On behalf of the authors

---

> ### Author Response · Authors · 2024-06-13
> **Thank you; we have updated the paper following your comments**
>
> Thank you so much again for your handling our paper.
>
> > Overall, this is a good paper overall, and can be accepted if the authors improve the paper's clarity
>
> Thank you so much for reading our paper and for your comments; we have answered to your comments and questions, and have revised some parts of the paper to improve its clarity.  And we have uploaded a paper of camera-ready format as instructed.
>
> > More discussion of the RDS model in Section 3.1. For example, it is stated that "RDS subsume many practical systems including Markov chains". Some examples of this would be helpful.
>
> Thank you for your comment; we have added additional explanations about how the additive noise systems and Markov chains could be represented by RDSs.  Also, we added an example of Markov chains and showed how it reduces to an RDS.
>
> > More explanation of the Koopman operator. The mathematical definition is in Definition 3.1, but some intuition would be helpful. For example, it is not clear whether such an operator K always exists or not.
>
> Thank you for the comment; we added the remark on "Choice of $\mathcal{H}$ and existence of $\mathscr{K}$" after defining the Koopman operator.  For your reference, we include the remark here as well: In an extreme (but useless) case, one could choose $\mathcal{H}$ to be a one dimensional space spanned by a constant function, and $\mathscr{K}$ can be defined.  In general, the properties of the Koopman operator depend on the choice of the space on which the operator is defined.  As more practical cases, if one employs a Gaussian RKHS for example, the only dynamics inducing bounded Koopman operators are those of affine (e.g., Ishikawa (2023); Ikeda et al. (2022)).  However, some practical algorithms have recently been proposed for an RKHS to approximate the eigenvalues of so-called "extended'' Koopman operator through some appropriate computations under certain conditions on the dynamics and on the RKHS (cf. Ishikawa et al. (2024)).
>
> > Example 3.1 can be expanded. How easy is it to compute the spectral radius of the Koopman operator (given that it is infinite dimensional)? What are the practical ways to compute it?
>
> Thank you for your comment; we have added extra explanations on the computations and approximations of spectra in Example 3.1.  We mention that studying the approximation techniques of spectra (including spectral radius) of an infinite-dimensional Koopman operator is an active topic of research, and is still a on-going topic.
>
> To obtain a theoretical guarantee that the computed spectral radius of a finite-dimensional matrix approximation well-approximates that of the target infinite-dimensional Koopman operator, the operator should be assumed compact.
>
> For your reference, we include the added sentences here as well: Assuming that the Koopman operator is defined over a finite-dimensional space, an eigenvalue of the operator corresponding to each eigenfunction can be given by that of the matrix realization of the operator.  In practice, one employs a finite-dimensional space even if it is not invariant under the Koopman operator; in such cases, there are several analyses that have recently been made for providing estimates of the spectra of the Koopman operator through computations over such a finite-dimensional space.  Interested readers may be referred to (Ishikawa et al., 2024;
> Colbrook \& Townsend, 2024; Colbrook et al., 2023) for example.  In particular, avoiding *spectral pollution*, which refers to the phenomena where discretizations of an infinite-dimensional operator to a finite matrix create spurious eigenvalues, and approximating the continuous spectra have been actively studied with some theoretical convergence guarantees for the approximated spectra.  In order to obtain an estimate of the spectral radius through computations on a matrix of finite rank, the Koopman operator may need to be compact (see the very recent work Akindji et al. (2024) for example).
>
> > Remark 3.1 is unclear to me. This is apparently an answer to one of the reviewers, but it may not be very clear for someone who hasn't read the discussion between the authors and the reviewers.
>
> Thank you for your comment; we have revised the remark and the figure.  We deleted the part of the figure showing the analogy to Fourier analysis and added some texts to clarify the comparison of the single-step cost formulation and the Koopman spectrum cost formulation.  Also, we have modified the remark to "Remarks on how the choice of $\mathcal{H}$ affects the Koopman spectrum cost" to explain the fact that the information carried by the Koopman operator varies depending on the choice of the function space.  We then improved the flow and clarity of the texts after the remark as well.

---

> ### Author Response · Authors · 2024-06-13
> **Additional responses**
>
> > $\mathbf{m}$ and $\mathbf{m^*}$ are used in Section 3.3 before being introduced much later in Appendix H.2. In general, I believe the discussion of Simulated experiments in Section 3.3 can be improved by adding more details.
>
> Thank you for your comment; we have moved the corresponding part in the appendix to the main text and adjusted it.  Also, we have expanded the explanations of each simulated experiment to clarify the detail.
>
> > Would you confirm that the minimum in the definition of regret is after the summation over t and not before that?
>
> Yes; we added a sentence there to clarify this.  It is partially because our algorithm is model based and the cost function that may vary over episodes is known, and the dynamics itself is invariant.

---

> > ### Comment · Action_Editor_HWAr · 2024-06-23
> >
> > Thank you for the revision of the paper and the summary here.